# Epigenetic Mechanisms of Plant Adaptation to Biotic and Abiotic Stresses

**DOI:** 10.3390/ijms21207457

**Published:** 2020-10-09

**Authors:** Vasily V. Ashapkin, Lyudmila I. Kutueva, Nadezhda I. Aleksandrushkina, Boris F. Vanyushin

**Affiliations:** Belozersky Institute of Physico-Chemical Biology, Lomonosov Moscow State University, 119234 Moscow, Russia; kutueva@genebee.msu.ru (L.I.K.); aleks@genebee.msu.ru (N.I.A.); vanyush@belozersky.msu.ru (B.F.V.)

**Keywords:** plant epigenetics, epigenetic variability, abiotic stress, biotic stress, environmental adaptation, gene expression, DNA methylation, chromatin, siRNA

## Abstract

Unlike animals, plants are immobile and could not actively escape the effects of aggressive environmental factors, such as pathogenic microorganisms, insect pests, parasitic plants, extreme temperatures, drought, and many others. To counteract these unfavorable encounters, plants have evolved very high phenotypic plasticity. In a rapidly changing environment, adaptive phenotypic changes often occur in time frames that are too short for the natural selection of adaptive mutations. Probably, some kind of epigenetic variability underlines environmental adaptation in these cases. Indeed, isogenic plants often have quite variable phenotypes in different habitats. There are examples of successful “invasions” of relatively small and genetically homogenous plant populations into entirely new habitats. The unique capability of quick environmental adaptation appears to be due to a high tendency to transmit epigenetic changes between plant generations. Multiple studies show that epigenetic memory serves as a mechanism of plant adaptation to a rapidly changing environment and, in particular, to aggressive biotic and abiotic stresses. In wild nature, this mechanism underlies, to a very significant extent, plant capability to live in different habitats and endure drastic environmental changes. In agriculture, a deep understanding of this mechanism could serve to elaborate more effective and safe approaches to plant protection.

## 1. Introduction

Plants live in a constantly changing environment that is often unfavorable or even hostile. As sessile organisms, plants cannot actively escape multiple aggressive encounters. Instead, they developed high phenotypic plasticity that includes rapid responses to aggressive environmental factors and adaptations to changing environments. Changes in gene expression underlie this phenotypic plasticity. Since gene expression is controlled by epigenetic marks, the epigenetic variation could be a key player in plant responses to stress factors and environmental adaptation.

The most thoroughly studied type of epigenetic phenomena in plants is DNA methylation [1,2]. A major part of methylated cytosine residues (m^5^C) in plants, like in animals, occurs in the symmetric CG sites. Unlike animals, plants also display significant methylation in the symmetric CHG sites and asymmetric CHH sites (here H is any nucleotide except G). All three methylation contexts are present in repeat and transposable element (TE) sequences, while the protein-coding gene sequences are mostly methylated at CG sites. The maintenance methylation of CG sites is carried out by DNA methyltransferase MET1 with the assistance of three VIM (VARIANT IN METHYLATION) family proteins, VIM1–VIM3 [3]. During DNA replication, this methylation complex recognizes and methylates with a high preference hemi-methylated CG sites in the daughter strands. A plant-specific DNA methyltransferase CMT3 (CHROMOMETHYLASE 3) is responsible for the maintenance methylation of symmetric CHG sites. It is probably involved also in the methylation of asymmetric CHH sites. Unlike MET1, CMT3 cannot recognize hemi-methylated sites and maintains CHG-specific methylation due to mutual stimulating substrate-level interactions between CMT3 and H3K9-specific histone methyltransferases (HMTs) [4]. DNA methylation de novo at CG, CHG, and CHH sites occurs mainly by the DOMAINS REARRANGED METHYLTRANSFERASE 2 (DRM2) [5]. Due to the asymmetric nature of CHH sites, their methylation is maintained by recurring methylation de novo. Recently, an alternative pathway dependent on CMT2 was found to participate in CHH methylation [6,7]. Similar to CMT3, CMT2 is targeted to methylated sites via histone H3K9 methylation marks. In contrast, DRM2 is targeted to its methylation sites due to the complementary interaction of 24-nt siRNAs (24-nucleotide small interfering RNAs) with sequences to be de novo methylated–the RNA-directed DNA methylation (RdDM) pathway [8,9].

In *Arabidopsis*, DNA could be actively demethylated via a base excision repair pathway involving the activity of dedicated m^5^C-specific glycosylase enzymes REPRESSOR OF SILENCING 1 (ROS1), DEMETER (DME), DEMETER-LIKE 2 (DML2), and DEMETER-LIKE 3 (DML3) [10]. It has been shown that DNA methylation status in multiple genome loci is the net result of their recurrent methylations by RdDM and demethylations by ROS1 [11].

Besides DNA methylation, plants and other eukaryotic organisms have another set of epigenetic marks–covalent modifications of various amino acid residues of histone proteins. Among the plethora of histone modifications, several types of methylation at lysine residues were best studied both in animals and plants [1,12]. In *Arabidopsis*, three types of H3K4 methylation marks (mono/di/tri-methylation–H3K4me1, H3K4me2, and H3K4me3) are found at gene bodies (H3K4me1) and promoters (H3K4me2 and H3K4me3) of actively transcribed genes [13]. H3K4me2/3 and m^5^C are mutually exclusive marks at the same promoter, while H3K4me1 could coexist with m^5^C along the gene bodies.

H3K27me3 shows a robust correlation with the repression of gene transcription at specific loci. Multiple genes are known to be regulated by this epigenetic mark during plant development, mostly independent of other epigenetic mechanisms [14]. Regulation of gene transcription by H3K27me3 marks is mediated by the Polycomb Repressive Complex 2 (PRC2) that includes an H3K27-specific HMT. The plant chromodomain protein LIKE HETEROCHROMATIN PROTEIN 1 (LHP1) binds to H3K27me3-containing genome loci and probably participates in mediating its regulatory effects [15].

H3K9me2 is a still another robustly repressive mark in plants. Unlike H3K37me3, this epigenetic mark is essentially heterochromatin-specific. As critical partners in non-CG DNA methylation by CMT3 and CMT2, H3K9me2 marks mostly colocalize with methylated CHG and CHH sites in repeat- and transposon-rich genome compartments [16].

Different kinds of small RNAs (sRNAs) in plants act via recognition of complementary sequences in mRNA or DNA, leading to posttranscriptional gene silencing (PTGS) due to the degradation of targeted mRNAs or inhibition of their translation (miRNAs and 21-22-nt siRNAs) or transcriptional gene silencing (TGS) due to DNA methylation via RdDM pathway (24-nt siRNAs) [17]. These sRNAs are produced via dsRNAs cleavage by different members of the DICER-like (DCL) family endoribonucleases and act as parts of the RNA-induced silencing complexes (RISCs) with the ARGONAUTE (AGO) family proteins.

This paper was not intended to be a comprehensive review of all works in the fascinating field dealing with epigenetic mechanisms in plant adaptation to various environmental stresses. That would be an unrealistic task in the frames of a single paper. Instead, we have tried to select recently published papers that contain robust data contributing essential knowledge to the epigenetic mechanisms of short-term and long-term plant adaptation. Of course, this selection reflects our personal view of the topic. Inevitably, many excellent papers remained unmentioned. We apologize to the authors for this omission. An interested reader could find further details in published review papers [18,19,20,21,22,23,24,25,26,27,28].

## 2. Epigenetic Responses to Stressful Factors

### 2.1. Abiotic Stress

Abiotic stresses mainly include extreme cold, heat shock, water deficit, excessive salinity, nutrient deficiencies, and heavy metal toxicity. To study pure (not caused by genetic factors) epigenetic variability, genetically uniform populations are usually used. When identical cloned lines of apomictic dandelion plants were exposed to various stresses (high salinity, low nutrients, defense response induced by jasmonic acid (JA) or salicylic acid (SA)), individual plants in all groups displayed significant variations in DNA methylation [29]. Similar though smaller variations were observed in the control (unexposed) group. These variations were mostly heritable (74–92%), and new variations often arose in daughter plants. These data show, first, that environmental stress increase epigenetic variability irrespective of genotype, and, second, that epigenetic differences occur both between plants exposed to different stresses and individual plants exposed to the same stress. Therefore, the epigenetic changes observed were mostly, if not exclusively, stochastic. Whether specific stresses could directly cause some of these epigenetic changes remain unknown.

#### 2.1.1. Cold Stress

Cold stress has profound effects on plant metabolism and gene expression. When exposed to low non-freezing temperatures, plants display increased tolerance to subsequent freezing temperatures–a phenomenon known as cold acclimation. Cold stress increases the levels of C-repeat binding factor family proteins (CBFs)–transcription factors that upregulate multiple cold-responsive (*COR*) effector genes [30]. PICKLE (PKL) is a subunit of the Mi-2/CHD3 subfamily of ATP-dependent chromatin remodelers that affects cold acclimation through the modulation of the CBF3 functional activity [31]. More than 600 genes were differentially expressed between the wild-type and *pkl* mutant plants after cold treatment, including the downregulation of *CBF3* and multiple CBF target genes, such as *RD29A*, *COR15A*, and *COR15B*. Since PKL is known to be involved in the RdDM [32] and H3K27me3 deposition [33] pathways, both H3K27me3 and DNA methylation could serve as memory marks for cold-induced freeze tolerance.

In *Arabidopsis*, WD40 repeat-containing protein HOS15 functions as a targeting protein in the ubiquitination-proteasome degradation pathway, while HISTONE DEACETYLASE 2C (HD2C) is one of its interacting partners [34]. Loss-of-function *hos15* mutant plants exhibit cold-sensitive phenotypes, irrespective of cold acclimation. In contrast, in *hd2c* mutants, freezing tolerance is comparable to that in the wild-type without cold acclimation and even better–with cold acclimation. Apparently, the histone H3 deacetylating activity of HD2C negatively regulates the expression of genes involved in cold acclimation, while HOS15 somehow counteracts this negative regulation. Consistent with this view, expression levels of *COR* genes (*COR15A*, *COR47*, and *RD29A*) are significantly reduced in *hos15* but increased in *hd2c* mutants compared with the wild-type upon cold treatment. Indeed, a HOS15-mediated proteasome degradation of HD2C at *COR* gene promoters was shown to occur upon cold treatment. Furthermore, HOS15 was found to assist in the binding of CBF proteins to the promoters of *COR* genes. This binding was significantly increased in cold-treated *hd2c* compared with wild-type plants, indicating that removal of HD2C by HOS15 is a prerequisite of CBF-binding in response to cold stress.

#### 2.1.2. Heat Stress

In heat shock (HS) response, heat shock transcription factors A1 (HsfA1s) serve as “master regulators” that activate multiple transcriptional networks [35]. Knockout mutants defective for these factors showed reduced induction of multiple HS-responsive genes and increased sensitivity to HS. Transcription of genes coding for essential HS-responsive transcription factors (TFs), such as DREB2A, HsfA2, HsfA7a, HsfBs, and MBF1C, is directly regulated by HsfA1s. Unlike animals, plants evolved extensive families of HS factors (HSFs) that differ in their expression patterns and functions. As master regulators, HsfA1s are indispensable in the HS response. However, their effects on the expression of HS-inducible genes are smaller than those of other HsfAs, such as HsfA2 and HsfA3, probably due to their own stringent regulation by post-translational modifications and interactions with other regulatory proteins.

In a study using a set of epigenetic mutants, Popova et al. [36] obtained the evidence that the RdDM pathway and the Rpd3-type histone deacetylase HDA6 play important and independent roles in basal heat tolerance. Moreover, the results of this study showed that nearby transposon sequences influence heat-dependent gene expression. HS induces the sustained accumulation of H3K9Ac and H3K4me3 on various heat shock protein genes [37]. Changes in histone modification and DNA methylation are directly relevant to both intergenerational and intragenerational forms of stress memory and, therefore, will be discussed in more detail in respective sections downstream.

#### 2.1.3. Salt Stress

By Na^+^ ion toxicity, hyperosmotic stress, and oxidative damage, high salinity greatly impacts plant growth and development. Evaluation of global DNA methylation levels in rice varieties largely different in salt tolerance found reduced DNA methylation after exposure to salt stress [38]. In leaves of the salt-tolerant variety Pokkali, the reduction in global DNA methylation was rapid and reached 70% hypomethylation. In contrast, in the salt-susceptible IR29 variety, the methylation loss was only 14% and non-statistically significant. In roots, the effect of salt stress on global DNA methylation was not statistically significant. These strikingly different changes in DNA methylation between the salt-tolerant Pokkali and salt-sensitive IR29 were correlated with distinct expression of the *DRM2* gene that was upregulated under the salt stress in IR29 but not Pokkali. In contrast, changes in the expression of two DNA demethylase genes were similar in both varieties; the *DNG701* gene showed a decrease after 1 h salt stress and an increase after 24 h salt stress, while the *DNG710* gene showed a gradual increase along salt stress.

Unlike the study above, another investigation of DNA methylation in rice cultivars upon salt stress found the most significant changes to occur in roots, while only slight changes were detected in leaves [39]. Whether this discrepancy was due to a different method to detect and quantify DNA methylation changes (methylation-sensitive amplified fragment length polymorphism–MSAP vs. immunological in [38]) or different rice lines used remains unknown. In general, the results indicated that salt stress-induced DNA methylation changes were mostly demethylation and that a substantial share of these DNA methylation changes was stable throughout the recovery period when the stress was removed. Four MSAP fragments were different between salinity-tolerant IL177-103 and salinity-sensitive IR64 under the control, stress, and recovery conditions. Genome sequences of IR64 and IL177-103 are very similar, and the sequences of the four polymorphic MSAP fragments appear to be identical. Thus, stable DNA methylation differences (epialleles) may be epigenetic markers responsible for phenotypic variations, including different salinity tolerance, between these closely related rice cultivars. The methylation pattern of MSAP fragments induced by salinity in root tissue was complicated. Some fragments displayed changed methylation that was stable during recovery; other fragments showed changed methylation that reverted to the control status after recovery. A few fragments were unchanged under salinity stress but changed after recovery. These polymorphic MSAP DNA fragments were associated with a wide range of gene functions, including stress responsiveness.

In plants, Ca^2+^-CALCINEURIN B-LIKE PROTEIN (CBL)-CBL INTERACTING PROTEIN KINASE (CIPK) complex participates in the regulation of cellular ion homeostasis [30]. High Na^+^, low K^+^, excess Mg^2+,^ and high pH cause cytosolic Ca^2+^ signals, which activate the SOS pathway, including SOS1 (Na^+^/H^+^ antiporter), AKT1 (K^+^ channel), Mg^2+^ transporter, and H^+^ ATPase. HIGH-AFFINITY K^+^ CHANNEL 1 (HKT1) mediates Na^+^ influx and, together with the SOS pathway, determines salinity tolerance in plants. In *Arabidopsis*, a putative siRNA target region at ~2.6 kb upstream of the *HKT1* gene start codon is heavily methylated in all sequence contexts in the wild-type plants [40]. In the *rdr2* mutant plants, deficient in small RNA biogenesis, CHG and CHH methylation of this region is significantly reduced, whereas CG methylation is unchanged. In the *met1* mutant, methylation in all sequence contexts is significantly reduced. Both mutations increase the *HKT1* expression in leaves, but only *met1* mutation increases the *HKT1* expression in roots relative to the wild-type plants. Furthermore, the DNA methylation-deficient mutant *met1* is hypersensitive to salt stress, while the *rdr2* mutant that lost non-CG methylation has normal salt sensitivity. Therefore, heavy methylation of the *HKT1* promoter in all sequence contexts inhibits transcription in leaves and roots, while non-CG methylation could serve to fine-tune the expression of *HKT1* in leaves, which may be essential in the long-term adaptation of plants to salt stress, but not in the short term salt tolerance. This DNA methylation-dependent regulation mechanism could be essential to balance *HKT1* expression between leaves and roots. In wild-type plants, the expression level of *HKT1* in roots is much higher than in leaves, while the transgenic plants that have reversed expression pattern of *HKT1* in roots and leaves (extremely high expression in leaves) show salt-hypersensitive phenotypes. The reversed *HKT1* expression pattern in these plants results in the rapid accumulation of Na^+^ in the leaves, which could explain their salt-hypersensitivity.

Besides DNA methylation, histone acetylation by HAT and deacetylation by HDAC complexes regulate plant adaptation to high-salinity stress [41].

In a study of long-term memory for salinity response, Sani et al. [42] showed that after a recovery period, *Arabidopsis* plants primed by exposure to mild salt stress displayed less salt uptake and higher drought tolerance than control plants. Specific changes in the H3K27me3 profiles occurred under the salt treatment and were maintained over a 10-day recovery period. The number of H3K27me3 islands increased from 6288 in non-primed to 7687 in primed plants. Despite this higher number, the overall genome coverage with H3K27me3 islands decreased in primed plants. An analysis of genome regions that differed in the levels of histone methylation between primed and non-primed plants showed that for H3K4me2 and H3K4me3, the vast majority of identified differential sites have higher methylation levels in the primed plants. By contrast, the vast majority of differential H3K27me3 sites showed lower methylation levels in the primed plants. About equal numbers of hypomethylated and hypermethylated sites were found for H3K9me3. These data indicate a more open chromatin configuration in primed plants without major changes in genome-wide histone modification profiles. For three genes, *HKT1*, *TEL1*, and *MYB75*, rapid and transient induction at mRNA level was found to be followed by a slower, long-lasting loss of H3K27me3. ChIP-qPCR analysis of nine selected genes showed that for five of them, the priming-induced loss of H3K27me3 was still present after a 10-d recovery period. The genome-wide profiles in the 10-d recovery plants reproduced the basic features discovered in the primed plants, including a larger number and lower genome coverage of H3K27me3 islands. Interestingly, in many cases, the gaps in H3K27me3 islands generated by the priming were progressively filled during recovery, probably due to the PRC2-mediated spreading of H3K27me3. Thus, priming-triggered demethylation of H3K27 might require active maintenance to prevent the fading of the molecular memory through H3K27me3 spreading. The lower shoot salt accumulation that was observed in primed plants upon the second salt treatment mimicked the phenotype of mutant plants over-expressing *HKT1*. In primed plants, increased *HKT1* mRNA levels were consistently observed after the second salt treatment at 10 d. Considering the observed loss of H3K27me3 at *HKT1* during the priming treatment and the HKT1 functional role as a root-specific Na^+^ transporter, the data obtained make HKT1 a prime candidate for explaining at least one of the priming physiological effects. The salt treatment was also found to change H3K27me3 and expression levels of three other genes. A plasma membrane aquaporin gene *PIP2E* was induced by salinity stress and still more induced in primed plants. *GH3.1* and *GH3.3* genes that encode auxin- and JA-conjugating enzymes, respectively, were also induced by salinity stress but displayed weaker induction in primed plants. These opposite priming effects on the *PIP2E* and *HKT1* (an increase in stress response) and *GH3.1* and *GH3.3* (a decrease in stress response) probably were accounted for by opposite effects of priming on H3K27me3 deposition, a decrease at *PIP2E* and *HKT1* and an increase at *GH3.1* and *GH3.3*. Thus, chromatin changes induced by salinity stress have no gross effects on constitutive gene expression but change the access of stress-inducible regulatory TFs to their target genes, thereby limiting any priming effects to reoccurring stress situations.

R2R3-MYB is the largest subfamily of the MYB family TFs known to regulate defense responses of plants. Several members of this subfamily were shown to participate in the abiotic stress responses [43]. In *Arabidopsis*, *MYB74* expression was strongly upregulated by salinity stress. In the *MYB74* overexpression transgenic lines, the expression of known stress marker genes, including *RD29B*, *RAB18*, and *RD20*, was also induced. All of these genes contain the conserved MYB recognition sites (TAACTG) in their promoters. Thus, MYB74 directly regulates the expression of the salinity stress genes. Significant DNA methylation in CG and CHH contexts and siRNA target sites were found in the *MYB74* promoter region. A noticeable reduction in m^5^C content was revealed by bisulfite sequencing in the *MYB74* promoter region when the wild-type plants were treated with salt. In the 200 bp promoter region approximately 500 bp upstream of the TIS, the percentage of CHH methylation was decreased by ~50%, that of CG methylation was decreased by ~10%, while no methylated CHG sites were found. The level of *MYB74* mRNA increased about eightfold under salt stress in a close correlation with the CHH demethylation. Five 24-nt siRNAs were predicted to target a narrow region (−603 to −477 bp) of the *MYB74* promoter. The accumulation of these 24-nt siRNAs was substantially reduced under salt stress. Therefore, a decrease in DNA methylation and induction of *MYB74* transcription under salt stress is probably due to the reduction in these 24-nt siRNAs.

#### 2.1.4. Water Deficit Stress

Most plants encounter water deficit stress many times across their lifespan. Multiple mechanisms help plants withstand these recurring drought encounters, including stress memory [24,28]. Abscisic acid (ABA) plays a vital role in regulating the activity of multiple drought stress-responsive genes. In *Arabidopsis*, repeated dehydration was found to upregulate several ABA-induced genes [44,45,46,47]. Moreover, the guard cell-specific memory maintained partially closed stomata across the recovery period [47].

The details of the drought stress response and resistance in plants are reasonably well studied [48]. Water deficit increases ABA production, which promotes the increased resistance to water deficit. The H3K4me3-specific methylase ATX1 stimulates the transcription of multiple genes involved in responses to biotic and abiotic stresses, including the drought stress [44]. The drought stress tolerance was accordingly diminished in the *atx1* mutant compared with the wild-type plants. This higher sensitivity of the *atx1* mutant plants to the water deficit was explained by more rapid transpiration by their leaves due to higher stomatal apertures. The ABA levels in *atx1* plants were only 40% of those in the wild-type plants. Of the four ABA biosynthetic genes, only *ABA3*, encoding an enzyme involved in the last step of ABA biosynthesis, and *NCED3*, supposedly the rate-limiting factor in ABA biosynthesis, showed diminished expression under dehydration stress in *atx1* compared with wild-type plants. ATX1 was shown to bind to a promoter region of the *NCED3* gene, while no such binding was observed for the *ABA3* gene. In accord with these results, the levels of H3K4me3 at the *NCED3* promoter region were increased by dehydration stress, and this increase was much higher in the wild-type than the *atx1* mutant plants. Four dehydration-inducible genes, *RD29A*, *RD29B*, *RD26*, and *ABF3*, were also induced by ABA. In the *atx1* mutant plants, the dehydration stress-induced transcription of these genes was significantly reduced relative to the wild-type plants, indicating that ATX1 participates in their regulation. Treatment with exogenous ABA restored the induced transcription of *RD29A* and *RD29B* to wild-type levels, whereas transcription of *RD26* and *ABF3* was partially restored. Of the four dehydration stress-responsive genes that were not dependent on ABA, *COR15A*, *ADH1*, and *CBF4* showed a strong dependence on ATX1, while *ABF2* showed only a modest dependence. The H3K4me3 levels at the representative dehydration stress-responsive genes from both groups showed a good correlation with their transcript levels, and genes downregulated in *atx1* plants showed reduced levels of H3K4me3.

The transcriptional responsiveness of genes induced by water deficit correlates with changed histone modifications and nucleosome density [41]. Intense dehydration stress leads to a more pronounced increase in H3K4me3 and H3K9ac and a decrease in nucleosome density on inducible genes compared with moderate dehydration. Thus, epigenetic responsiveness appears to depend on the intensity of the stress. During recovery from stress, H3K9ac rapidly decreased, and RNA polymerase II was removed from the drought stress-upregulated genes, while H3K4me3 was decreasing gradually upon rehydration.

In *Arabidopsis*, LHP1 is a component of the repressive complex PRC1 that binds to H3K27me3 marks via its chromodomain. The binding of LHP1 to ABA-responsive genes *ANAC019*, *ANAC055*, and *VSP1* and their H3K27me3 levels decreased after ABA treatment [49]. Thus, LHP1 contributes to their repression via increased H3K27me3 marks. ANAC019 and ANAC055 are known as positive regulator TFs of drought tolerance. The *lhp1* mutant plants showed increased ABA sensitivity and significantly higher tolerance to a prolonged drought period than the wild-type plants. Therefore, LHP1 negatively regulates ABA-mediated responses to drought, probably via increased H3K27me3 at *ANAC019* and *ANAC055*.

The cumulative effect of multigenerational drought stress on genome-wide DNA methylation was studied by an MSAP method in drought-sensitive (II-32B) and drought-resistant (Huhan-3) rice cultivars that were grown under drought stress for six successive generations [50]. II-32B showed more differentially methylated loci (DMLs) between F_0_ and F_6_ generations and between normal and drought treatments (~13% of total 3070 loci) compared with Huhan-3 (~1.8% of total 4739 loci). Among the 402 DMLs in II-32B, 254 showed no difference between normal and drought conditions in F_0_ and accordingly were considered to be unaffected by drought stress. In contrast, 112 and 36 DMLs became re-methylated or de-methylated after drought stress in F_0_, respectively. Most of these loci (74.1% and 77.8%, respectably, which account for ~27.6% of total 402 DMLs) still retained their changed methylation status after drought treatment in F_6_. Therefore, these loci could be directionally affected by drought stress, as they tend to change methylation similar in both F_0_ and F_6_. Huhan-3 has only 84 DMLs, 30 became re-methylated, and 21 de-methylated after drought stress in F_0_. Of these, 23 were still re-methylated, and 18 still de-methylated in F_6_ after drought stress. Therefore, ~48.8% of a total of 84 DMLs were directionally induced by drought stress in Huhan-3. Remarkably, in II-32B, 8 of 112 DMLs that became re-methylated after drought stress in F_0_ retained the re-methylated status in F_6_ without drought stress. Similarly, among the 36 DMLs that became de-methylated after drought stress in F_0_, 21 loci retained the de-methylated status in F_6_ without drought stress. Collectively, these sites accounted for ~7.2% of the total 402 DMLs. The stability of their methylation status across six generations means that these loci might be stably inherited between generations. In Huhan-3, there were 24 (80% of 30 re-methylated DMLs) and 16 (~76.2% of 21 de-methylated DMLs) loci that showed transgenerational inheritance, accounting for ~47.6% of total 84 DMLs. Therefore, a larger proportion of DMLs was inheritable in the drought-tolerant rice cultivar. These findings could have important implications in understanding the place of epigenetic variation in plant evolution.

In *Populus trichocarpa*, drought stress-induced changes in DNA methylation were studied by the high-resolution WGBS method [51]. Genome-wide, m5C content appeared to be significantly higher in drought stress-exposed than control plants (10.04% and 7.75% of total cytosines, respectively). The transcriptome sequencing analysis showed a general positive correlation between the expression levels of expressed genes and their methylation levels, while heavy methylation often led to gene silencing. In drought-stressed plants, ~7400 genes showed an increase, and ~10,300 genes showed a decrease in methylation and transcription compared with control plants. Decreased DNA methylation and expression after drought treatment were found in 1156 genes encoding TFs, including *MYB*, *AP2*, *WRKY*, *NAC*, and *bHLH* families. Increased DNA methylation and expression after drought stress were found in 690 genes coding for TFs, mostly of *C3H*, *PHD*, *MYB*, *ARF*, and *bZIP* families. Thus, changed DNA methylation could be a regulatory mechanism affecting the gene expression response to drought stress at the genome-wide scale.

#### 2.1.5. Multiple Stresses

Sixty annual clones of a stress-tolerant poplar genotype *Populus simonii* “QL9” were used in a comparative study of epigenetic effects of four abiotic stress treatments (salinity, osmotic, heat, and cold) [52]. The total DNA methylation levels significantly increased after 3 h of treatment for all four stresses; the effect of HS was significantly higher than of the other stresses. In the HS, the cytosine methylation levels reached a maximum at 6 h and remained unchanged after that. In contrast, in three other stress treatments, the cytosine methylation levels gradually increased until 24 h. At 24 h, the cytosine methylation levels under osmotic and cold stress treatments were higher than under HS. In the genome-wide DNA methylation patterns, 39121 MSAP fragments appeared to be differentially methylated between control and stressed plants, relative levels of both mCG and mCHG being highest under osmotic stress and lowest in the control group. Heat and osmotic stress had the maximal number of common methylated sites, while cold stress had minimal numbers of overlapping methylated sites with heat and salt stress. A total of ~1400 functionally diverse DMRs were found, including 104 TF genes, 23 protein modification genes, 68 protein degradation genes, 39 receptor kinase genes, 18 calcium regulation genes, eight G-protein genes, and others. The patterns of stress-specific methylated fragments were different between the four abiotic stresses. Among the MSAP fragments that showed no homology to protein-coding genes, 35 were mapped to miRNAs and lncRNAs. These ncRNAs and their putative target genes showed different patterns of stress-responsive expression. In the control group, 87.9% of methylation sites were unchanged at 1 and 2 months, while 64.8% of these sites were still unchanged 6 months after stress treatment. In contrast, only 35.1% of de novo methylated sites were still present at 1 month. After 2 and 6 months, only 23.8% and 15.3% of such sites, respectively, were maintained. Of the stress-demethylated sites, 28.9%, 17.7%, and 11.3% were maintained for 1, 2, and 6 months, respectively. Following cold and osmotic stress treatments, 18.7% and 17.6% of de novo methylated sites, respectively, were maintained for 6 months, significantly more than after heat and salt stresses. Among the 1376 DMRs, 373, 289, and 164 DMRs were detected at 1, 2, and 6 months after the abiotic stress. Of 14 DNA methyltransferase and DNA demethylase genes, only *DNMT2* was significantly induced under salinity stress, while the expression of other genes did not change. It is worth to be noted that in plants, DNMT2 probably serves not as a DNA methyltransferase but rather as a C5-tRNA-methyltransferase [53]. Under osmotic stress, transcript levels for methylation-related genes increased, including *DRM2*, *MET1*, and *DDM1*, but the expression of demethylation-related genes was unaffected. Under HS, the transcript levels of *DDM1.1*, *DRM2*, *MET1.3*, and *MET1.1* increased, while those of *DDM1.2*, *MET1.2*, *ROS1*, and *DME* decreased. Under cold stress, multiple genes in both categories were induced, including *CMT3.1*, *CMT3.2*, *DRM2*, *MET1.1*, *DNMT2*, *ROS1*, *DME1*, and *DME3*. Thus, different abiotic stresses had different effects on the transcription of genes related to DNA methylation.

#### 2.1.6. Nutrient Deficits Stress

Very much like any other environmental factor, nutrients are perceived by multiple signaling pathways assisting in plant adaptation to their fluctuating availability in the soil [54]. High nitrogen (N) represses the expression of a root nitrogen transporter, NRT2.1, via a negative feedback loop mediated by HNI9 (High Nitrogen Insensitive 9)—a critical factor in the deposition of repressive H3K27me3 marks at the *NRT2.1* gene [55].

The iron homeostasis is negatively regulated by the PRMT5-mediated symmetric dimethylation of the histone H4 third arginine residue (H4R3sme2) at several genes of the *bHLH* family subgroup Ib [56]. Indeed, in the PRMT5 deficient mutant plants, higher iron accumulation in shoots and greater tolerance to iron deficiency were observed relative to the wild-type plants. In *Arabidopsis*, a mutation in *GCN5* (General Control Nonrepressed protein5) was found to impair the iron translocation from the root to the shoot [57]. The expression of *GCN5* reached a maximal level at 3 d of iron-deficiency treatment, being upregulated by more than fivefold over the control. A total of 879 putative GCN5-regulated genes potentially involved in iron homeostasis were identified by transcriptome sequencing in wild-type and *gcn5* mutant plants grown either with sufficient or deficient iron supply. Significant shares of these genes were implicated in responses to abiotic stress. Five genes related to iron transport, *FRD3*, *EXO70H2*, *MLP329*, *BOR1*, and *CRK25*, appeared to be direct targets of GCN5. Consistent with the known function of GCN5 as a histone acetyltransferase, the H3K9ac and H3K14ac and mRNA levels of these five genes under iron-deficiency conditions were significantly decreased in *gcn5* mutant compared with the wild-type plants. The *AL6* (*Alfin Like 6*) gene plays an important role in root hair formation induced by phosphate starvation and several other processes related to cellular phosphate homeostasis [58]. Since AL6 is known to be a PHD finger reader protein of H3K4me3 epigenetic marks, these data indicate a possible role of H3K4me3 and other chromatin marks in plant adaptation to phosphate deficit. In two WGBS studies in *Arabidopsis* and rice, the phosphate (Pi) starvation was found to induce multiple changes in DNA methylation [59,60]. In rice, it led to transient changes in DNA methylation, especially the hypermethylation of TEs near Pi-stress-induced genes [59]. Unexpectedly, changes in transcription preceded changes in DNA methylation in apparent contradiction of the common view of the causal relationship between DNA methylation and transcription. No intergenerational transmission and limited intragenerational stability of induced changes in DNA methylation were observed. Relative to rice, Pi starvation induced a relatively small number of changes in DNA methylation in *Arabidopsis*, possibly due to a lower number of TEs.

In striking contrast, other authors reported that Pi starvation in *Arabidopsis* results in extensive remodeling of global DNA methylation and that local changes in DNA methylation often correlate with changes in the transcription of nearby genes [60]. Moreover, DNA methyltransferase genes *MET1*, *DRM1*, *DRM2*, and *CMT3*, and DNA demethylase genes, *ROS1* and *DML2*, were all upregulated by Pi starvation. The only exception was *DML3* that showed downregulation. A global analysis of differentially methylated C residues (DMCs) showed that the changes in DNA methylation induced by Pi starvation affected both CG and non-CG sites. About 23–37% of all DMCs were specific to gene-related regions, suggesting that a wide range of genes are regulated by DNA methylation during *Arabidopsis* response to Pi starvation. By analysis of mutant plants defective in different DNA methyltransferases, both CG and non-CG methylation were shown to be required for the correct Pi starvation response. The differential cytosine methylation near the Pi-responsive motif sequences was shown to correlate with gene expression, suggesting that methylation of these regulatory elements could affect the binding of respective TFs and thereby control the transcription.

In a zinc (Zn) deficiency tolerant *Arabidopsis* genotype Sf-2, a prolonged Zn deficiency treatment upregulated 189 and downregulated 430 genes more than twofold [61]. The Zn transporter family genes *ZIP1*, *ZIP3*, *ZIP4*, *ZIP5*, and *IRT3*, the nicotianamine synthase genes *NAS2* and *NAS4*, the heavy metal ATPase gene *HMA2*, and the purple acid phosphatase gene *PAP27* were among the most robustly induced genes. The only downregulated gene in this list of the most robustly regulated genes was *TERMINAL FLOWERING 1* (*TFL1*). Genome-wide changes of DNA methylation were observed upon prolonged Zn deficiency treatment, leading to the upregulation of some Zn deficiency-responsive genes. Hypo- and hypermethylated DMRs were identified, but hypomethylated DMRs dominated, especially in the non-CG context. Most CG-DMRs were found in the gene bodies, TEs, and intergenic regions, while non-CG-DMRs occurred predominantly in TEs, but also intergenic regions. Most genes of the robustly Zn-regulated core set were distant from any DMRs. The majority of responsive genes had unaltered methylation patterns. In the *Arabidopsis* genome, 83 genes contain Zn deficiency-responsive promoter motif RTGTCGACAY, including *ZIP4*, *ZIP5*, *IRT3*, and two defensin-like genes *AT1G34047* and *AT4G11393*. The methylation level of both cytosines in this promoter motif was consistently low across all genes. Globally, no correlation between changes in DNA methylation and transcription was observed upon prolonged Zn deficiency.

In maize roots, 4807 differentially expressed genes (DEGs) were identified between the control and the prolonged Zn deficiency-treated plants, about equal numbers of them being up- and downregulated [62]. Unlike at short or mild Zn deficiency treatments, several genes of Zn uptake systems, *ZIP3*, *ZIP4*, *ZIP5*, *ZIP7*, and *ZIP8*, were substantially upregulated. Additionally, *NAS* genes, especially the abundant NAS5, were consistently upregulated upon Zn deficiency. NAS enzymes synthesize nicotianamine, which is involved in the translocation of heavy metals, including zinc and iron, between organs. The downregulated gene set was significantly enriched for genes encoding enzymes of the oxidative stress response, such as peroxidases and superoxide dismutases. Several genes encoding the maintenance DNA methylation enzymes (MET1, CMT3, DDM1) were also downregulated, while the *ROS1* gene encoding a DNA demethylating enzyme was upregulated. An RRBS analysis of DNA methylation showed a massive loss of methylation in the CG and CHG contexts in the Zn deficiency group. In the control group, 26.6% of CGs and 18.7% of CHGs were methylated, while in the Zn-deficiency group, these methylation levels were about twofold less (13.2% and 9.7%, respectively). The CHH sites showed a very low methylation level in control (1.26%) and slightly further reduced methylation in Zn-deficiency samples (1.06%). In total, 2762 DMRs were identified between the two groups, most of them in the CG context. Consistently with decreased overall methylation, most DMRs were hypomethylated, though a few hypermethylated DMRs were also identified. Most DMRs in both contexts were associated with TEs. Eight percent of genes with DMRs in the CG context were also differentially expressed. In contrast, only a single gene with DMR in the CHG context was differentially expressed. Differentially methylated promoters/gene bodies and differentially methylated TEs were found about equally among up- and downregulated genes. Apparently, gene expression can be either repressed or stimulated by DNA demethylation.

### 2.2. Biotic Stress

#### 2.2.1. Viruses

Among the first evidence for the epigenetic regulation of plant tolerance towards biotic factors was the control of viral virulence via PTGS [63]. Upon infection by RNA viruses, plants recognize viral double-strand RNA molecules, inducing their degradation into siRNAs by DCL2 and DCL4. Another mechanism, TGS, provides for a more permanent defense against DNA viruses via RdDM. PTGS and TGS function not only in protection against virus infections but also in the regulation of plants’ own genes. Unlike animals, plants do not have adaptive immune systems. Instead, they evolved a most complex RNA-based system of gene regulation and protection against foreign nucleic acids. Most plant viruses have a single-stranded RNA (ssRNA) genome. The immune response triggered by RNA viruses often leads to the degradation of their genomes into siRNAs. Since DCL family endoribonucleases act on dsRNA, the genomes of ssRNA viruses are first converted into dsRNA molecules by RNA-dependent RNA polymerases. Recently, it was shown that m^6^A-specific methylation of the RNA genome in the alfalfa mosaic virus (AMV) controls the infection in *Arabidopsis* [64]. The *Arabidopsis* protein ALKBH9B (At2g17970) was shown to possess a demethylase activity that removes m^6^A from single-stranded RNA molecules in vitro and accumulates in the cytoplasm in siRNA bodies, suggesting that ALKBH9B is an m^6^A demethylase involved in mRNA silencing and/or mRNA decay processes. ALKBH9B was shown to affect the infectivity of AMV but not of cucumber mosaic virus (CMV), correlating with the ability of ALKBH9B to bind or not to their coat proteins. Suppression of ALKBH9B increased the relative abundance of m^6^A in the AMV genome and impaired its systemic invasion of the plant, while it was without effect on CMV infection.

#### 2.2.2. Microbes

The role of DNA methylation in plant immunity has been exhaustively studied [63,65]. The first layer of active defense, known as pathogen-associated molecular pattern (PAMP)-triggered immunity (PTI), relies on the perception of PAMPs or microbe-associated molecular patterns (MAMPs) by pattern-recognition receptors (PRRs). PAMP perception is followed by the activation of immune responses, which results in basal immunity. To override the plant defense, pathogens produce special effector molecules that damp PTI. As a counter-counter defense, these pathogen effectors could be perceived by disease resistance proteins, often resulting in a potent immune response–effector-triggered immunity (ETI). Activation of both PTI and ETI involves massive changes in gene expression regulated by epigenetic mechanisms.

In *Arabidopsis*, infection by a bacterial pathogen *Pseudomonas syringae* pv. *tomato (Pst)* elicits a basal defense response that is suppressed by bacterial virulence factors. Plants recognize some of these factors and activate defense and hormonal pathways, including the upregulation of SA signaling. Unexpectedly, mutant plants defective in CG (*met1*) or non-CG (*drm1 drm2 cmt3*–*ddc*) methylation were markedly resistant to *Pst* infection [66]. In both mutants, transcript levels were unchanged for most genes, but numerous genes were misexpressed (>10-fold change was observed for about 2000 genes in *met1* and about 1300 genes in *ddc*). Exposure to pathogen changed expression levels of multiple pathogen-responsive genes in both mutants compared with the infected wild-type plants. In total, these data show that changed DNA methylation of some genes could be an important mechanism of plant defense. Interestingly, CG- and CHG-specific methylation was similarly altered in plants exposed to SA or avirulent or virulent *Pst* strains. In contrast, changes in CHH-specific methylation were unique to *Pst* infection, suggesting that CHH methylation, but not CG or CHG methylation, respond differently to different stresses. Unexpectedly, mostly hypomethylated (77%) DMCs were observed under SA treatment, whereas mostly hypermethylated (89%) were observed in response to avirulent *Pst* infection. A Gene Ontology (GO) analysis of DMRs induced by virulent *Pst* or SA showed a strong enrichment for genes involved in plant defense, while avirulent bacteria induced DMRs associated with genes involved in transcriptional regulation and, to a lesser extent, defense responses. Most of these genes were misregulated in *met1* and *ddc* mutant plants. Therefore, DNA methylation is an essential part of the transcriptional control of these genes under stress conditions.

Flagellin Sensing 2 (FLS2) is a well-characterized plant PRR that senses the bacterial flagellin-derived peptide flg22, leading to changed expression of multiple genes [67]. The flg22 treatment was shown to reactivate TEs and other well-characterized RdDM targets, suggesting that it inhibits TGS. Indeed, a significant downregulation of the key components of the RdDM pathway occurred at 3 h and 6 h after flg22 treatment, which correlated with the upregulation of the early defense gene *Flg22-induced Receptor-Kinase 1* (*FRK1*). The majority of the RdDM components returned to normal levels at 9 h concomitant with induction of the late defense gene *Pathogenesis-related gene 1* (*PR1*). Progressive flg22-triggered demethylation occurred at the retroelement *AtSN1* and *ONSEN*’s LTR regions, primarily in the CHH context. This DNA demethylation preceded the activation of *AtSN1* and *ONSEN*, suggesting that transcriptional activation may be caused by demethylation. Moderately increased bacterial growth was observed in *ros1* mutants infected with *Pst* strain DC3000, supporting the role of active DNA demethylation in antibacterial resistance. The SA-dependent defense response was decreased in *ros1* mutant plants, as indicated by an attenuated induction of *PR1* by flg22. In sharp contrast, mutants defective in RdDM displayed lower bacterial growth, even more so in mutants defective both for RdDM and maintenance DNA methylation, consistent with active PR1 expression that mimics the flg22-induced expression observed in the wild-type plants. Plant NOD-like receptors (NLRs) are key immune receptors whose overexpression triggers a constitutive hypersensitive response (HR) and/or *PR1* induction. The HR-like phenotype and enhanced *PR1* expression were observed in *met1 nrpd2* double mutant plants, suggesting that some *NLRs* might be directly controlled by RdDM. In flg22-elicited wild-type leaves, 55 *NLRs* were upregulated more than twofold. Among these genes, 15 had closely associated repeats/siRNA clusters, and six of these *NLRs* were expressed at higher levels in *met1 nrpd2* mutant than wild-type plants. The *Resistance Methylated Gene 1* (*RMG1*) was upregulated by flg22 in wild-type plants and untreated *met1 nrpd2* mutant plants compared with untreated wild-type plants. In RdDM-defective mutant plants, it was induced by flg22 earlier and more sustainedly compared with wild-type plants. *RMG1* encodes a nucleotide-binding site leucine-rich repeat (NB-LRR) protein with a Toll/interleukin-1 receptor (TIR) domain. Its promoter region contains a *helitron*-related repeat sequence targeted by siRNAs and heavily methylated in all sequence contexts. DNA methylation of the downstream proximal promoter region was low in wild-type plants but drastically increased in the *ros1* mutant. Basal expression and upregulation of *RMG1* by flg22 were completely abolished in *ros1* mutant plants. Thus, the *RMG1* gene is methylated via the RdDM pathway and demethylated by ROS1. Both its basal expression and flg22-triggered upregulation depend on the DNA demethylation by ROS1.

The innate immunity responses in plants are often short-term but can elicit the acquired immunity state that manifests itself as “priming” of inducible defenses [68]. Primed plants respond faster and/or stronger to recurring defense stimuli. Priming could be induced by microbes, as in pathogen-induced systemic acquired resistance (SAR). Other priming states can be triggered by chemicals, such as β-aminobutyric acid (BABA). Some priming states are relatively short-term and disappear within a few days, while others are long-lasting and can even be transmitted between plant generations. The priming of SA-dependent immunity is long-lasting and transgenerationally inheritable, suggesting the involvement of epigenetic mechanisms.

The possible involvement of DNA methylation in resistance against biotrophic pathogens was studied in *Arabidopsis* DNA methylation mutants infected with obligate biotrophic downy mildew oomycete *Hyaloperonospora arabidopsidis* (*Hpa*) [68]. Microscopic examination of *Hpa* colonies showed that two mutants defective in RdDM, *nrpe1* and *drd1*, had statistically significant increased resistance. The *cmt3* mutant also showed enhanced resistance relative to the wild-type plants but lesser than *nrpe1* and *drd1*. The *ddm1* mutant showed the strongest level of resistance amongst all genotypes tested. In contrast, the DNA-hypermethylated mutant *ros1* was significantly more susceptible to *Hpa* than the wild-type plants. Since the resistance to *Hpa* depends on SA-dependent defenses, expression of the SA-inducible *PR1* marker gene was quantified at 48 and 72 h post-infection. Consistent with respective resistance phenotypes, *nrpe1* mutant plants displayed stronger induction of the *PR1* gene, while *ros1* mutant plants showed decreased *PR1* induction compared with the wild-type plants. Therefore, DNA hypomethylation primes SA-dependent defense against *Hpa*, whereas DNA hypermethylation suppresses it.

The mutants in the RdDM pathway showed decreased resistance to the necrotrophic fungus *Plectosphaerella cucumerina* associated with repressed responses of JA-inducible defense genes [68]. At 6 days postinfection, the *nrpe1* mutant plants developed larger necrotic lesions, while *ros1* mutant plants displayed significantly smaller lesions than wild-type plants. Similar results were obtained for a different necrotrophic fungus, *A. brassicicola*. Thus, DNA hypermethylation in the *ros1* mutant increases disease resistance to necrotrophic fungi. Basal resistance against *P. cucumerina* and *A. brassicicola* partially relies on JA-dependent defenses. The *nrpe1* mutant showed significantly lower and/or delayed JA induction of defense genes *PDF1.2* and *VSP2* relative to wild-type plants. Surprisingly, despite the higher resistance, the *ros1* mutant plants also showed diminished induction of *PDF1.2* and *VSP2* by JA. Thus, increased resistance of *ros1* to necrotrophic fungi was not based on primed responsiveness of JA-inducible gene expression. At three days after SAR induction in *Arabidopsis* by infiltration with an avirulent strain of *Pst* DC3000, wild-type plants displayed a statistically significant reduction in *Hpa* sensitivity compared with untreated control plants. In similarly infected *nrpe1* mutant plants, SAR was statistically non-significant, probably due to the elevated basal resistance. Notably, the *ros1* mutant plants were fully capable of mounting a statistically significant SAR response against *Hpa* infection, indicating that ROS1-dependent DNA demethylation does not play a role in establishing within-generation SAR. When the wild-type, *nrpe1*, and *ros1* plants were inoculated three times with increasing doses of a virulent strain of *Pst* DC3000, 3-wk-old F_1_ seedlings from wild-type plants showed increased basal resistance in comparison to progeny from control non-inoculated plants. By contrast, no statistically significant difference in *Hpa* resistance was observed between treated and non-treated progenies of *nrpe1* plants. Levels of resistance in the non-treated progeny of *nrpe1* plants were statistically similar to that of *Pst* DC3000-treated progeny of wild-type plants. Thus, reduced DNA methylation in *nrpe1* plants could mimic transgenerational acquired resistance (TAR). Like in the *nrpe1* mutant, *Pst* DC3000-treated and non-treated progenies from *ros1* plants did not show a difference in *Hpa* resistance. However, non-treated progeny from *ros1* displayed increased susceptibility compared with both *Pst* DC3000-treated and non-treated progenies of wild-type plants, indicating that the lack of TAR in *ros1* is due to this mutant’s inability to transmit and/or express transgenerational acquired immunity. Of the 967 *Hpa*-responsive genes, 49% were affected by mutations in *NRPE1* and/or *ROS1*. Thus, nearly half of the pathogenesis-related transcriptome in *Arabidopsis* is controlled by NRPE1- and ROS1-dependent DNA methylation-demethylation.

In a search for *Hpa* resistant epigenomes, 123 epiRIL lines of *Arabidopsis* were analyzed [69]. Four epigenetic quantitative trait loci (epiQTLs) were identified, accounting for 60% of the variation in disease resistance. Important, none of these epiQTLs were associated with growth impairment or make plants more susceptible to other infections or environmental stresses. Higher resistance in the *Hpa*-resistant epiRILs was associated with the genome-wide priming of defense-related genes. None of these epiRILs displayed increased basal transcription of SA-dependent defense marker gene *PR1*, but all of them showed increased induction of *PR1* at 48–72 h post-infection with *Hpa* compared with wild-type plants. Besides, compared with wild-type plants, EpiRILs showed an increased response of callose deposition—an essentially SA-independent pathogen-inducible defense mechanism. Hence, most *Hpa*-resistant epiRILs are primed to activate differentially regulated defense responses, explaining the lack of major costs on growth and compatibility with other types of stress resistance. A large set of genes involved in SA-dependent and SA-independent defensive responses showed augmented *Hpa*-induced expression in the epiRILs compared with wild-type plants. Collectively, these results indicate that the increased resistance of the epiRILs is based on the priming of *Hpa*-inducible defense genes. Interestingly, only a small share of these genes appeared to be located within the borders of the epiQTL intervals. Additionally, a relatively large proportion of defense-related genes was shared between all four epiRILs, though these epiRILs carried different combinations of the four epiQTLs. Therefore, the majority of *Hpa*-inducible genes that showed increased expression in the more resistant epiRILs are trans-regulated by DNA methylation at the four epiQTLs.

In a similar experimental setting, 16 epiQTLs were detected that affect resistance of *Arabidopsis* epiRILs to clubroot–a *Brassica* disease caused by *Plasmodiophora brassicae* [70]. Six epiQTLs were mapped close to the clubroot resistance genes and QTLs. Thus, both allelic and epiallelic variations could interact with the environment, leading to variable clubroot resistance.

In potato, priming with BABA increased resistance to the oomycete pathogen *Phytophthora infestans*, the causal agent of late blight disease [71]. The first unstressed generation of the BABA-primed parent plants showed increased resistance to the *P. infestans*, probably due to the upregulation of SA-responsive genes. During the early priming phase, a bivalent histone mark configuration, H3K4me2 and H3K27me3, was observed on the SAR regulator genes *NPR1* (Non-expressor of *PR* genes) and *SNI1* (Suppressor of *NPR1*, Inducible). This readily switchable between active and silent states chromatin configuration increased responsiveness of the *PR1* and *PR2* genes, thus contributing to intergenerational stress memory. Maintaining BABA-primed defense memory did not depend on histone acetylation until the plants were triggered with the *P. infestans*. After that, the rapid and boosted expression of *PR* genes probably required HAT activity both in parents (F_0_) and progeny (F_1_). BABA treatment resulted in fourfold downregulation of the pathogen gene *Pitef1* at 48 hpi compared with unprimed inoculated plants [72]. The progeny plants (F_1_) of BABA-primed potato (F_0_) showed 2–2.5-fold downregulation of *Pitef1* at 48 hpi compared with the progeny of unprimed plants. The expression of the *MET1* gene was unaffected both after the BABA treatment and pathogen infection. In contrast, the *CMT3* and *DRM2* genes were upregulated about 7-fold and 18-fold, respectively, at 3 h upon BABA treatment. Thus, DNA methylation by CMT3 and DRM2 may be essential in BABA priming. The DNA demethylase gene *ROS1* was highly upregulated at 3 h (60-fold) and 6 h (20-fold); in the following hours, its expression rapidly diminished. At the later phase of priming (24 to 48 h), 12-fold upregulation of *DML2* was observed, suggesting active demethylation. Collectively, these results suggested that BABA induces DNA methylation followed by active removal of the m^5^C marks, and changed DNA methylation status of the target genomic regions may underlie the long-lasting priming memory. However, no correlation between promoter DNA methylation of SA-dependent genes and their expression was observed in BABA-primed plants. On the other hand, the promoter of the key potato resistance gene *R3a* showed a rapid increase in methylation level at 6 h after BABA treatment correlated with the observed downregulation of its expression. In turn, after *P. infestans* inoculation, the methylation level of the *R3a* promoter drastically diminished, while its expression significantly increased. When tested in various types of defense responses, a robust negative correlation between the levels of the *R3a* promoter methylation and its expression was observed. Furthermore, the offspring of BABA-primed plants exhibited promoter hypomethylation and a high level of the *R3a* gene expression compared with the unprimed potato plants in good accord with the intergenerational resistance to *P. infestans.*

A reverse genetic screen was used to identify HMTs that regulate PTI [73]. For PTI activation to the necrotrophic fungi *Botrytis cinerea*, the *Arabidopsis* endogenous peptide 1 (pep1) was used; pep1 represents a damage-associated molecular pattern that is recognized by PEPR1 and PEPR2 receptors to activate plant immunity, including resistance to necrotrophic fungi. In *Arabidopsis*, the genes encoding HMT proteins are named *SDG* (*SET Domain Group*) genes. Mutants of 10 *SDG* genes responsive to infection were tested for fungal resistance to *B. cinerea* and/or PTI. SDG25 and SDG8 were shown to regulate pep1-triggered immunity to fungal infection. The *sdg8* and *sdg25* mutant plants displayed increased sensitivity to *B. cinerea* infection before and after pretreatment with pep1 and decreased pep1-triggered immunity to *B. cinerea* compared with wild-type plants. Thus, SDG8 and SDG25 actively contribute to pep1-triggered immunity to fungal infection. Both *SDG25* and *SDG8* genes were significantly upregulated by *B*. *cinerea* infection and pep1 treatment. The *Arabidopsis* mutant *hub1* impaired in the H2Bubn-specific E3 ligase also completely lacked pep1-triggered immunity to *B. cinerea*. Thus, both histone methylation and ubiquitination are required for that kind of immunity. A similar analysis of the mutant sensitivity to *Pst* DC3000 infection showed that SDG25 and SDG8 are also implicated in plant immunity to bacterial pathogens. Further analyses showed that SDG8 broadly contributes to the ETI, while SDG25 has only a limited contribution specific to certain effectors. Both SDG8 and SDG25 appeared to play essential roles in SAR. Consistent with the known functions of HMTs as epigenetic transcription regulators, in the *sdg8 sdg25* double mutant plants, 6063 genes failed to be induced by *B. cinerea* compared wild-type plants. The loss of these SDGs significantly impacted the *Arabidopsis* transcriptome, with roughly 25% of the normally infection-responsive genes losing their responsiveness. Besides, in the *sdg8 sdg25* double mutant plants, 4941 genes were upregulated to a greater extent than in the wild-type plants upon *B. cinerea* inoculation, suggesting these genes to be negatively regulated by SDGs. Multiple defense genes implicated in bacterial and fungal resistance were directly affected by these SDGs. Globally, SDG8 affected both H3K36 and H3K4 methylation, while SDG25 affected mostly H3K4 methylation. H3K4me3, H3K36me2, and H3K36me3 levels were significantly diminished in the *sdg8 sdg25* double mutant plants, consistent with the observed effects on gene expression and defense responses.

In tomato, inoculation with *B. cinerea* was shown to highly upregulate genes *DES*, *DOX1*, and *LoxD* that encode key enzymes in the oxylipin pathway, and *WRKY75* that encodes a stress-responsive TF [74]. An increase in H3K4me3 and H3K9ac levels in all the pathogen-induced genes occurred concomitantly with their activation. These same genes were also induced in response to *Pst* DC3000. An increase in H3K4me3 and H3K9ac was also observed with this pathogen, though, along *DES* and *DOX1*, it was significantly less than with *B. cinerea*. However, *WRKY75* showed a significant increase in both histone marks along the gene.

#### 2.2.3. Pests

The soybean cyst nematode (SCN; *Heterodera glycines*) penetrates soybean roots to induce the formation of a multinucleated feeding site–the syncytium. Induced changes in genome methylation were studied by comparative WGBS analysis of infected and non-infected soybean roots [75]. Average methylation levels were similar between the SCN-infected and control samples. In the CG context, 718 hypermethylated regions (hyperDMRs) and 1408 hypomethylated regions (hypoDMRs) were identified in the infected compared with non-infected roots. Similarly, in the CHG context, 1142 hyperDMRs and 2074 hypoDMRs were identified, while in the CHH context, 605 hyperDMRs and 1210 hypoDMRs were identified. Thus, hypomethylation was the prevalent trend in all sequence contexts. In total, 703 and 1346 genes were found to be hyper- and hypomethylated, respectively. Only 25 genes were hypermethylated in more than one context. Both hyper- and hypomethylation of various parts were found in 45 genes. In a set of 24 randomly selected DMR-containing genes, 21 genes changed their expression in response to SCN infection. Differential methylation induced by SCN infection had various effects on gene expression. Among genes that were differentially methylated in SCN-infected plants, 93 hyperDMR genes and 193 hypoDMR genes overlapped with the 6903 genes that change the expression in response to SCN infection.

In a follow-up study, the effects of SCN infection on the genome-wide DNA methylation profiles were studied in SCN-resistant and SCN-susceptible near-isogenic soybean lines (NILs) [76]. In contrast to the high genetic similarity between highly SCN-susceptible (S) and highly SCN-resistant (R) lines, their DNA methylomes were found to be very different. Under stringent criteria (≥50% methylation difference, FDR = 0.01), 21852 unique DMRs between S and R plants were identified, including 4180 that overlapped with 3666 protein-coding genes and 11,211 DMRs that overlapped with 6033 TEs. CG-DMRs were mostly found in gene bodies and, to a much lesser extent, in gene promoters and 5′- and 3′-UTRs. About 70% of CHG-DMRs were found in gene bodies. TE-associated DMRs were mostly mapped to LTR retrotransposons. A total of 948 DEGs were identified, 587 of them upregulated and 361 downregulated in S relative to R plants. GO analysis showed significant enrichment for functions related to wounding and defense responses, membrane disassembly, and intracellular signal transduction, indicating that under non-infected conditions, differential gene expression between the NILs may underly their different capabilities to respond to SCN infection. In S plants, SCN infection led to reduced DNA methylation levels over protein-coding genes in all sequence contexts, whereas in R plants, an opposite effect was observed. In response to SCN infection, 50040 DMRs were identified in the S plants relative to non-infected control, 7584 of them overlapped with 6252 protein-coding genes, 28100—with TEs. The number of DMRs in the R plants was much lower. A total of 5080 DMRs were identified in the R infected samples compared with the R control samples, 1296 of them overlapped with 1293 protein-coding genes, and 2356—with TEs. Thus, a massive DNA methylation reprogramming occurred only during the susceptible interaction with pathogen. A total of 1668 DEGs were identified in S plants, and only 112 in R plants at 5 d post-SCN infection relative to non-infected controls. GO analysis of these DEG sets revealed significant enrichment for plant responses to oxidative stress, chemical stimulus, and oxidation-reduction in R plants, while in S plants, significant enrichment was found for responses to stimuli, and phytohormone signaling, including ethylene, SA, JA, and ABA. Only one gene was common between sets of 1293 differentially methylated genes (DMGs) and the 112 DEGs identified in the R plants upon SCN infection. When these same DMGs were compared with syncytium DEGs, 188 genes overlapped. In a similar analysis in S plants, 123 genes were common between DMGs and DEGs identified upon SCN infection. Furthermore, 50 DEGs that contained differentially methylated TEs in their gene body or promoters were also found, resulting in a unique list of 147 differentially expressed DMGs. When these same DMGs were compared with syncytium DEGs, 756 genes overlapped. Collectively, these results show that DNA methylation changes during SCN infection have an impact on gene expression. When DNA methylomes of non-infected S and R plants were compared with methylomes of their parental lines, 58 DMRs were identified between NILs that were inherited from parental lines. Of these, 38 DMRs were hypomethylated in NIL-R and its SCN-resistant parental line Fowler, but hypermethylated in NIL-S and its SCN-susceptible parental line Anand. The 20 DMRs were hypermethylated in Fowler and NIL-R, but hypomethylated in Anand and NIL-S. These 58 DMRs overlapped with 57 protein-coding genes, of which four are differentially expressed in SCN-induced soybean syncytium. Interestingly, 56 DMRs unique to NIL-S were identified. Gain or loss of methylation in these DMRs was the opposite of that detected in the parental lines and NIL-R. These 56 DMRs overlapped with 55 protein-coding genes, including 9 of the previously identified syncytium DEGs. Thus, respective permanent changes in methylation could play a role in soybean–SCN interaction.

#### 2.2.4. Parasitic Plants

Parasitic plants use specialized organs, haustoria, to penetrate the host plant tissues and extract nutrients and water for their own growth and reproduction. Besides nutrients and water, haustoria serve as channels for transports of signaling molecules, protein, DNA, and RNA [77]. To identify host and parasite mobile transcriptomes, cDNA libraries were derived from the host (*Arabidopsis* or tomato) stem parts that were free of the parasite *Cuscuta pentagona* tissue, from the *Cuscuta* stem parts that were free of the host tissue, and from sites of the parasite attachment that contained tissues of both plants [78]. The Illumina reads from each library were assigned to host or parasite transcriptomes to estimate RNA movement between the species. *Arabidopsis* reads in parasite tissue accounted for 1.1% of total reads, whereas host stems contained 0.6% of *Cuscuta* reads. A similar pattern was found in the tomato–*Cuscuta* association, though somewhat lower rates of transfer were estimated to occur. The greatest number of mobile transcripts originated from *Arabidopsis* hosts. About 45% (9518) of the expressed *Arabidopsis* transcripts were detected in *Cuscuta*. In contrast, only 1.6% (347) of the expressed tomato transcripts were detected in *Cuscuta*. Concerning movement from parasite to host, 24% (8655) of the expressed *Cuscuta* mRNAs were detected in *Arabidopsis*, while only 0.8% (288) of the expressed *Cuscuta* mRNAs were detected in tomato. Thus, the volumes of mRNA traffic between *Cuscuta* and the two hosts were consistent in both directions, suggesting that haustorial selectivity is regulated by the host plant. The reason for differences in haustorial selectivity between *Arabidopsis* and tomato remained unknown. It probably reflects some active mechanisms to resist infection present in tomato, such as the secretion of defensive compounds at the infection site [79]. One of the factors affecting the mobility of different mRNAs was their abundance in the cells near the host–parasite boundary [78]. However, many transcripts with similar abundance showed differing mobility. The abundance of most mobile *Arabidopsis* transcripts in the parasite tissue was about one-hundredth of that in the interface tissues, indicating similar dynamics of movement. However, some host RNAs occurred in parasite at levels nearly equal to those in the interface. Host mRNAs mostly disappeared from *Cuscuta* within several hours, but some of them were detected over long distances in parasite stems up to ~20 cm from the haustorial connection [80]. Whether mobile mRNAs have a function remains unclear, though their delayed degradation in foreign tissues suggests some functional significance.

Some sRNAs that move between parasite and host plants are known to function trans-specifically [81]. Recently, sRNA expression in *C. campestris* grown on *A. thaliana* was studied by deep sequencing [82]. Relative to the parasite stem, 76 *C. campestris* sRNA species were significantly upregulated in the host-parasite interface, including 43 miRNAs. One of these miRNAs was a member of the conserved *miR164* family, while the others had no significant sequence similarity to known miRNA loci, and none of them aligned perfectly with the *A. thaliana* genome. Six *Arabidopsis* mRNAs were predicted to be targets of movable *Cuscuta* miRNA. No endogenous *C. campestris* mRNA targets were found to any of the induced miRNAs, suggesting that these miRNAs have evolved to avoid targeting the *C. campestris* own transcripts. Instead, these miRNAs may function to target mRNAs of the host plant. Indeed, five of these putative target mRNAs were significantly downregulated in parasitized compared with control stems. SEOR1 and AFB3 mRNAs were among the six putative targets of mobile *Cuscuta* miRNA. Mutant plants *seor1* and *afb3* showed significantly increased susceptibility to *C. campestris*. Therefore, the mobile miRNA of *C. campestris* targets host mRNAs in a biologically relevant way to counteract the host defensive mechanism. Overall, the results suggest that *C. campestris* trans-species miRNAs function to change host gene expression in a way beneficial to the parasite. Collectively, the data described indicates that epigenetic interactions shape the dynamics of the “arms race” between host and parasite plants.

## 3. Short-Term Epigenetic Memory (Priming) of Stress

In natural environments, plants continuously experience unfavorable encounters. In evolution, they elaborate specific adaptive mechanisms to overcome various kinds of environmental stress and retain the stress response information for some time after the stress encounters are over. It was shown that stress factors induce alterations in the epigenetic status of stress-response genes that could still be present for some time after recovery or even in the progeny [32,42,45,47,83]. This kind of information used by plants to respond faster or stronger at repeated exposure to the same stress was named stress priming or stress memory [45,84]. *Arabidopsis* plants subjected to several cycles of dehydration/water recovery retained more water than plants experiencing dehydration stress for the first time [45]. Moreover, these treatments affected gene expression in two different ways. Some genes were expressed at similar levels during each stress treatment, while other genes significantly increased their expression at repeated treatments relative to the first treatment. Accordingly, genes in the second category could be referred to as “stress memory genes,” while the genes in the first category are just stress-responsive (“non-memory”) genes. Two distinct marks were found at the memory genes during the recovery period: high levels of H3K4me3 and stalled form of RNA polymerase II–Ser5P PolII (phosphorylated at the serine 5th). In contrast, on stress-responsive non-memory genes, these two marks dynamically increased at stress treatment and then decreased to basal levels during the recovery period. At the memory genes, H3K4me3 and Ser5P Pol II persisted for as long as the transcriptional memory lasted. A comprehensive RNA-Seq analysis of the *Arabidopsis* transcriptomes prior to dehydration stress and after the first and third stress exposures revealed a high diversity of memory-type responses [46]. In total, 6579 genes were significantly affected by the first stress (S1), compared with normally watered plants (W), about equal numbers of them upregulated and downregulated. Furthermore, 1963 dehydration-responsive genes displaying significantly different levels of transcripts after the third stress (S3) compared with S1 and therefore were referred to as “memory genes.” Of these 1963 memory genes, 362 genes were upregulated in S1 and upregulated to higher levels in S3. Similarly, 310 memory genes were downregulated in S1 and still more downregulated in S3. Interestingly, 434 genes were downregulated in S1 but transcribed in S3 at higher levels than in S1. Last but not least, 857 memory genes were upregulated in S1 but expressed in S3 at lower levels than in S1. Thus, the genes in two latter groups “revised” their response in subsequent stress: after robustly responding in S1, these genes show weaker/no responses in S3. Accordingly, these genes were referred to as “revised response” memory genes. GO analysis showed that about a quarter of the memory genes in the first category were implicated in cold/heat acclimation and responses to salt and ABA. In the second category, the highest enrichment was found for genes encoding ribosomal, chloroplast, and photosynthesis proteins. No enrichment for any particular functions was detected in the third category. The last group was enriched for genes implicated in signaling pathways, such as ABA, ethylene, auxin, gibberellic acid, JA, and SA. Since H3K4me3 marks were found to be associated with stress-memory genes as a “memory mark” [45], in follow-up work, the possible association of this epigenetic mark was studied in the “revised response” memory genes that showed robust induction at the first stress treatment (S1) but lower or absent induction at repeated stress treatments [85]. Consistent with the transcription levels, high H3K4me3 levels were found in S1, but were low before the first stress-treatment (W), after watered recovery (R1), and after the second stress exposure (S2). Therefore, in this category of stress memory genes, H3K4me3 does not serve as a memory mark. No significant changes were found in H3K27me3 levels for all tested genes, irrespective of high (S1) or low (W, R1, and S2) transcription. Thus, H3K27me3 also does not serve as a stress memory mark for these genes. Furthermore, a similar study on five stress memory genes in the “superinduced” (induced in S1 and still higher induced in S2) genes showed that the H3K27me3 levels along these genes were practically constant throughout all phases of the treatment cycle, irrespective of their transcription levels [86]. Surprisingly, in S2, H3K27me3 levels were practically unchanged from the pre-stressed state (W) despite super-activated transcription. Thus, H3K27me3 is not a memory-mark at the drought stress memory genes.

A ChIP-qPCR study of chromatin modifications in *Arabidopsis* plants exposed to priming heat stress (HS) revealed the involvement of the heat stress-induced gene *HSP22.0* in heat stress memory [37]. During the 2 days following HS, *HSP22.0* transcripts remained increased, while those of *HSP70* returned to the basal level. Both genes were strongly enriched for H3K9ac at 4 h after HS. This enrichment declined rapidly at *HSP70*, where it was no longer significant at 28 h and undetectable at 52 h. In contrast, *HSP22.0* remained significantly enriched at 52 h. A moderate (threefold) enrichment for H3K4me3 was observed at *HSP70* at 4 h after HS, then it declined over the next 2 days and returned to basal levels by 52 h. In striking contrast, *HSP22.0* showed high (up to 75-fold) enrichment for H3K4me3 that remained highly pronounced at 52 h. Interestingly, no changes in H3K4me2 were detected at *HSP70*, while an enrichment at *HSP22.0* was observed at the later time points (28, 52 h) but not at 4 h. Thus, *HSP22.0* remained significantly enriched for H3K4me3 and H3K4me2 at the later time points when the gene expression and H3K9ac levels had declined. Therefore H3K4me3 and H3K4me2 marks at *HSP22.0* may represent transcriptional memory of the priming HS.

## 4. Transgenerationally Inherited Epigenetic Memory of Stress

In addition to the memory during priming, epigenetic changes could transmit between plant generations. Multiple cases of such transgenerationally inheritable epigenetic changes (epimutations), both naturally arisen and artificially induced, were described [26]. The formation of stress-induced transgenerational memory obviously should benefit the plant progeny to achieve a better stress resistance [87,88].

One of the most widely accepted roles of DNA methylation in plant genomes is the control of transposon activity [2,89]. In *Arabidopsis*, HS transiently destabilized TGS at the constitutive heterochromatin loci rich in transposable elements, but TGS was re-established by 24 h after HS [90]. A notable exception was a small retrotransposon family *ONSEN* (Japanese “hot spring”), which retained a high level of transcription. In the genome of the Columbia accession *Arabidopsis*, *ONSEN* consists of eight members. *ONSEN* transcripts were detected in HS plants directly after the stress treatment and for up to 3 days of recovery. No such transcripts were detected in control (non-stressed) plants. HS-induced accumulation of *ONSEN* transcripts was significantly higher in plant mutant for components of 24-nt siRNA biogenesis. During the recovery period, *ONSEN* transcripts gradually decreased and were undetectable after 10 days in wild-type and all mutant plants. Therefore, siRNA-mediated regulation appears to restrict the *ONSEN* transcript levels after HS but is not involved in their resilencing during the recovery period. In the DNA of *Arabidopsis* plants subjected to HS, a significant increase in the *ONSEN* copy number was detected. In *nrpd1* and other siRNA biogenesis mutant plants, the number of *ONSEN* DNA copies exceeded 500 compared with the maximal number of about 50 in wild type plants. These copy numbers gradually decreased during the subsequent 20–30 days recovery period in both wild-type and *nrpd1* plants, eventually returning to the initial number of the Columbia accession. No chromosomal integration events were detected. No ONSEN transpositions were detected in the offspring of non-stressed *nrpd1* and wild-type plants, as well as of HS-treated wild-type plants. Surprisingly, frequent transpositions were observed in the progeny of *nrpd1* mutant plants subjected to HS. The patterns of new *ONSEN* insertions were quite variable even between the sibling progeny of the same plant. The insertion sites of new *ONSEN* copies showed a genome-wide distribution but with a clear preference for transcribed gene regions. Interestingly, a gene in the Columbia accession harboring a natural insertion of *ONSEN* showed a highly increased HS-induced transcriptional activation compared with the Zurich accession, where *ONSEN* is absent at this location. Even more intriguingly, in the second generation of *nrpd1* HS-treated plants, two loci harboring new *ONSEN* insertions showed heat responsiveness that was absent in wild-type or the first-generation *nrpd1* plants. Therefore, after an induced burst of *ONSEN* transposition, different subsets of genes in various progeny plants acquire new regulatory properties.

In a follow-up study, epigenetic regulation of the *ONSEN* activity was studied in more detail [91]. Temperature shifts from 24 h at 6 °C to 24 h at either 21, 27, or 32 °C did not activate *ONSEN* transcription and did not change the copy number of *ONSEN* DNA in wild-type and *nrpd1* mutant plants. The *ONSEN* transcript levels and DNA copy numbers increased only in plants subjected to a temperature shift from 6 °C to 37 °C. In *nrpd1* mutant plants, HS-induced *ONSEN* transcript level was about 28-fold higher, and the *ONSEN* DNA copy number was about 85-fold higher than in the wild type plants. Therefore, *ONSEN* activation is regulated by a heat-sensitive factor with an activation threshold at around 37 °C and a siRNA-mediated pathway. In the wild type, HS-treatment of the parental plants did not affect the HS-induced *ONSEN* transcription in the progeny. In contrast, in the *nrpd1* mutant plants, HS-treatment of the parental plants resulted in an about 55-fold increase in HS-induced *ONSEN* transcription and much higher copy numbers of *ONSEN* DNA in the progeny. Furthermore, a high frequency of new transpositions was detected in the HS-treated F_3_ generation progeny of the HS-treated *nrpd1* F_2_ generation progeny of the HS-treated F_1_ generation parental plants. Therefore, the HS-induced transposition occurred in each generation. In total, the results described show that the newly inserted copies of *ONSEN* in each generation were activated by the HS and re-silenced without HS. A high level of *ONSEN* transcripts was shown to be an important prerequisite of successful transgenerational transpositions [92]. 

The detailed molecular mechanism of the *ONSEN* HS regulation was elucidated by Cavrak et al. [93]. When 3-wk old plants of *Arabidopsis* were subjected to a prolonged (up to 30 h) HS, *ONSEN* RNA was first detectable at 6 h after the onset of HS, and its amount increased to the highest level at 24 h, remaining high until 30 h—the endpoint of the HS treatment. Small amounts of linear extrachromosomal *ONSEN* DNA were first detectable at 12 h. The highest levels were achieved at 30 h after HS onset, corresponding to approximately five times more than genomic *ONSEN* DNA. Therefore, the production of the linear extrachromosomal *ONSEN* copies follows the transcriptional activation in a short time, probably needed to complete the reverse transcription of *ONSEN* RNA. Sequencing analysis of isolated extrachromosomal *ONSEN* DNA copies showed 57% of them derived from the three most evolutionary young *ONSEN* loci (*ONSEN 1-3*). Three other *ONSEN* loci (*ONSEN 4-6*) contributed 23% of the extrachromosomal *ONSEN* DNA copies, and two of the eight loci (*ONSEN 7, 8*) were not represented at all.

In *Arabidopsis*, transposon mobility is known to be restricted by DNA methylation at multiple CHG and CHH sites imposed by DNA methyltransferases DRM2 (RdDM pathway) and CMT3 CMT2 (H3K9me pathway) [6,7]. However, the LTR sequences of active *ONSEN* copies contain only CHH sites [93]. This finding is unexpected, considering that CHH methylation per se cannot ensure a stable transposon silencing. Rather such silencing is achieved by redundant CHG and CHH methylation [7]. A bisulfite sequencing analysis of 5′LTR sequences of several *ONSEN* elements before and after HS showed that at ambient temperature, the two most evolutionary young loci (*ONSEN 1* and *2*) have high CHH methylation profiles across their 5′LTRs [93]. Unexpectedly, *ONSEN 8* that was not activated appeared to be substantially less methylated. After HS, *ONSEN 8* became more methylated, while *ONSEN 1* and *2* lost methylation at several sites. In the triple mutant *ddc*, the levels of CHH methylation at the *ONSEN* LTRs were significantly reduced even in non-stressed plants. Nevertheless, no *ONSEN* transcription or extrachromosomal *ONSEN* DNA production was detected. After HS, the CHH methylation in the triple mutant was virtually unchanged, but the levels of *ONSEN* mRNA and extrachromosomal DNA were significantly increased. Sequencing analysis of *ONSEN* extrachromosomal DNA showed an even more dominant representation of activated *ONSEN 1* and *2* in mutant compared with wild-type plants. Therefore, the reduction of DNA methylation at the retrotransposon promoter does not per se activate its transcription but promotes the activation by HS. In the 5′LTR sequences of all *ONSEN* loci, a heat response element (HRE) with the consensus sequence nTTCnnGAAn was found. HREs are known to bind trimers of heat shock factors (HSFs) to induce downstream target genes. In *Arabidopsis*, all four HSFA1-type TFs are constitutively expressed, while HSFA2 is not detectable at ambient temperatures but strongly and stably expressed under HS. Both *HSFA2* and *ONSEN* transcription showed the same dependence on the individual HSFA1 factors. The recombinant HSFA2 protein was found to bind the *ONSEN* LTR probe with high specificity. In the *hsfa2* mutant plants subjected to HS, *ONSEN* activation was severely reduced, although not completely abolished. Thus, HSFA2 is probably a major, but not the only HSF involved in the HS-induced *ONSEN* activation. Transgenic analysis of the *HSFA2* and *ONSEN* promoters by fusion with reporter genes showed that both are preferentially active in dividing meristem cells. Thus, by recruiting an HSFA2-binding HRE into its promoter, *ONSEN* exploits an important defense mechanism of its host to ensure its own successful multiplication. On the other hand, new *ONSEN* insertions render downstream genes HS-responsive, which might be useful as a new adaptive feature.

In *Arabidopsis*, some transposable elements (TEs) are located within introns of transcribed genes and marked by repressive epigenetic modifications [94]. Plants evolved a specific mechanism to override the inhibitory effect of heterochromatic domains within introns. Two factors play critical roles in this process. One of them is ENHANCED DOWNY MILDEW 2 (EDM2) that recognizes H3K9me2 marks via its PHD domains. Another is INCREASE IN BONSAI METHYLATION 2 (IBM2) that binds to H3K9me2-containing nucleosomes via an RNA-recognition motif (RRM) and a Bromo-Adjacent Homology (BAH) domain. IBM2 controls *ONSEN 1* copy inside the intron of an F-box protein. The stress tolerance in *ONSEN*-integrated progenies of the HS-treated *nrpd1* mutant plants was investigated by seeding on a medium containing ABA [95]. When tested in wild-type plants, ABA elicits response mimicking the result of environmental stress responses. Among the *ONSEN*-integrated progenies of the HS-treated *nrpd1* mutant plants, two plants showed an ABA-insensitive phenotype suggesting the stress tolerance. In one of them (13–7), new *ONSEN* insertions occurred in 24 loci, including 20 located within genes, including 14 within exons. In the other (19–4), new insertions were found in 21 loci, including 13 within genes, including eight within exons. In the 13–7 ABA-insensitive line, one of the new *ONSEN* insertions occurred in the first intron of the *ABSCISIC ACID-INSENSITIVE 5* (*ABI5*) gene. In the 19–4 line, a new *ONSEN* insertion was detected in the exon of the *ABSCISIC ACID-INSENSITIVE 4* (*ABI4*) gene. Consistent with ABI4 and ABI5 functions as TFs that regulate ABA-inducible genes, most genes known to be differentially expressed under ABA stress exhibited low ABA-sensitivity in mutant lines. Indeed, 13–7 and 19–4 plants displayed ABA-insensitive phenotypes at germination similar to those in *abi5* and *abi4* mutants. In addition, significant shares of genes directly regulated by ABI4 and ABI5 were suppressed in 19–4 and 13–7, respectively.

Collectively, the results of the HS-induced *ONSEN* transposition studies show that even a transitory relaxation of epigenetic control induced by stress factors could lead to genetic and epigenetic variability, potentially increasing the chances of new adaptive phenotypes.

When *Arabidopsis* and tomato plants were exposed to caterpillar herbivory, JA, or mechanical damage, transgenerational priming of JA-mediated defense responses was observed [96]. The growth of the corn earworm *Helicoverpa zea* on the progeny of treated tomato plants was diminished by about 40% relative to control plants. In *Arabidopsis*, feeding of parental generation plants to *Pieris rapae* (white cabbage butterfly) reduced caterpillar weight in progeny by about 40%. JA treatment had a smaller effect (−27%), while mechanical damage was without effect. When the progeny of *P. rapae*-treated parents was exposed to three other lepidopteran herbivores, the crucifer-specific diamondback moth (*Plutella xylostella*) and two species with broader host ranges, the cabbage looper (*Trichoplusia ni*) and the beet armyworm (*Spodoptera exigua*), besides the *P. rapae* itself, only *S. exigua* showed reduced growth. This memory of stress priming persisted in *Arabidopsis* for two generations and disappeared in the third generation. *Arabidopsis* mutants that are deficient in JA perception (*coronatine insensitive1*) or the biogenesis of siRNAs (*dcl2 dcl3 dcl4* triple mutant or *nrpd2a nrpd2b* double mutant) did not exhibit inherited resistance. Collectively, these data suggest that induced changes in DNA methylation via the RdDM pathway serve as epigenetic memory marks, providing for a phenotypically plastic mechanism of enhanced defense across generations.

The role of epigenetic mechanisms in plant long-term adaptation to the adverse natural environment was studied in two rice epimutation accumulation (epi-MA) lines obtained by drought imposition through 11 successive generations of drought-tolerant and drought-sensitive rice varieties [87]. The frequency and distribution of induced epimutations were studied in WGBS DNA methylomes of 32 progeny plants. In both lines, progeny plants of the eleventh generation (F_11_) displayed higher tolerance to osmotic stress than their respective parental plants of the original generation (F_0_). DNA methylomes were analyzed in both lines in three stressed generations (F_0_, F_11_, and F_10_), drought-treated and well-watered plants in each of them, and a recovery generation F_10R1_ derived from stressed F_10_ by cultivation in well-watered conditions for one more generation. Similar patterns of epimutations were induced by drought in F_0_ and successive generations, indicating a long-term impact of drought on DNA methylation patterns. The influence of drought on DNA methylation was not random but showed some directionality because the epimutations in offspring generations were clustered together. In successive generations of stressed sibling plants, F_11_ and F_10R1_ were more similar to F_10_ than to F_0_, again indicating some directionality in the stress-induced accumulation of epimutations. Nevertheless, the number of epimutations did not increase linearly with time. The number of epimutations obtained through a single generation between F_10_ and F_11_ was much larger than expected from the total epimutations accumulated between F_0_ and F_11_. This apparent discrepancy is probably due to many epimutations not being stably inheritable between generations. The percentages of recurring epimutations between drought-treated and untreated plants in different generations were much higher than expected from random distribution across the genome, suggesting that drought-induced epimutations occur in “hot spots.” Very high shares (>45%) of drought-induced epimutations maintained their changed statuses in advanced generations suggesting that these epimutations could be considered epigenetic markers that participate in the long-term adaptation of plants to drought conditions.

Whether the frequently occurring heavy metal-contamination of soil should be regarded as environmental stress that has transgenerationally inheritable effects on DNA methylation was studied in rice [88,97]. Treatment of 10-day old rice seedlings with the heavy metals, Cu^2+^, Cd^2+^, Cr^3+^, and Hg^2+^, significantly inhibited shoot and root development and changed DNA methylation patterns in four of the six TEs and 11 of the 12 protein-encoding genes that were analyzed [97]. Being essential trace elements, Cu^2+^ and Cr^3+^ caused gross changes in DNA methylation only at high concentrations (1 mM), while Cd^2+^ and Hg^2+^, being nonessential heavy metals, induced DNA methylation changes at 50 μm. In Hg^2+^-treated plants (F_0_ generation), changed methylation was observed in most studied sequences (66.7%). In the progeny (F_1_ generation) of a single Hg^2+^-treated F_0_ plant, three types of methylation patterns were detected: ‘‘inheritance’’ of the modified methylation patterns of the parental F_0_ plant, ‘‘new patterns’’ superimposed on the modified patterns of the F_0_ plant, and ‘‘reversion’’ to the original patterns of the untreated control plants. Each type of methylation patterns occurred at variable frequencies among the eight genes that were tested, but overall, the ‘‘new pattern’’ was the predominant type (45% to 95%). Thus, further DNA methylation modifications in the same direction (mostly CHG-specific hypomethylation) occurred in the F_1_ generation. No changes were observed in the F_1_ progeny of control (non-stressed) plants, proving the changes observed in the progeny of stressed plants not to be due to spontaneous epigenetic variation. Further transgenerational changes in DNA methylation were studied in the progeny (F_2_ generation) of two F_1_ plants, one (#4) that showed predominantly new patterns, and another (#11) that showed all three types of patterns for different genes. For most genes that were tested, the new patterns which had been detected in F_1_ were stably inherited in F_2_ progenies. Furthermore, the stable inheritance of these methylation patterns was observed in the F_3_ generation plants. Therefore, changes in DNA methylation patterns induced by the heavy metal stress, after becoming homozygous for the epiallelic state, are stably inheritable under unstressed conditions. Moreover, the sensitivity tests showed that progeny plants of the Hg^2+^-stressed plants developed heritable enhanced tolerance to the same stress. In a follow-up study, the authors aimed to determine whether different classes of genes have common or specific responses to heavy metal stress [88]. To this end, 18 functionally diverse genes were chosen, of which two (*Tos17* and *Osr42*) were formerly found to display changed methylation in response to the heavy metal stress, seven were randomly chosen, and nine (*HMA1*–*HMA9*) comprised the P_1B_ subfamily of Heavy Metal-transporting P-type ATPases (HMAs) involved in the uptake and transport of heavy metals in plants. F_0_ generation plants that showed the most noticeable changes in DNA methylation patterns under heavy metal treatments were selected for the gene expression analysis. Two TEs, *Tos17* and *Osr42*, showed upregulated expression under all or three of the four heavy metal treatments. Of the seven single-copy genes tested, five showed transcriptional upregulation in all heavy metal treated plants. Of the nine rice *HMA*s, seven showed significant upregulation under at least one of the four heavy metal treatments. The possible inheritance of changed gene expression patterns was studied in the F_1_ progeny seedlings of one F_0_ Hg^2+^-stressed plant grown under normal non-stress conditions. All fourteen genes that were tested showed transcriptional changes in the parental F_0_ plant. The expression patterns in the F_1_ progeny plants were classified into three types: “inheritance” of the Hg^2+^-treated F_0_ pattern, “reversion” to the un-stressed F_0_ pattern, and the differential pattern that was further sub-divided into transgenerational memory (further upregulation) and other. For both TEs (*Tos17* and *Osr42*), the F_1_ progeny exhibited inheritance of the F_0_ pattern or its further upregulation. Two of the four single-copy genes, specifically *Homeobox* and *HSP70*, exhibited stable inheritance of the F_0_ pattern in most F_1_ plants (75% and 87.5%, respectively) and its reversion to the un-stressed F_0_ pattern in the rest. One gene, *YF25*, showed inheritance, reversal, and novel pattern in the F_1_ progeny at frequencies of 25, 37.5, and 37.5%, respectively. For *SNF-FZ14*, further upregulation was observed in 75% of the F_1_ plants and the inheritance in the remaining 25%. Of the eight *HMA* genes tested, *HMA1* showed further upregulation in 100% progeny; *HMA2* showed 50% inheritance and 50% reversal; *HMA4* showed inheritance in 37.5% and reversal in 62.5%; *HMA5* showed inheritance, reversal, and further upregulation in 50, 25, and 25%, respectively; *HMA6*, *HMA7*, and *HMA8* showed inheritance in 25, 12.5, and 62.5%, and further upregulation in 75, 87.5, and 37.5%, respectively; and *HMA9* showed inheritance in 100% progeny. In total, all genes tested showed some inheritance of the Hg^2+^ stress-induced gene expression patterns, but to a variable extent. Among the F_2_ progeny of one F_1_ plant that showed all three expression patterns for several of the tested genes, 36.6% inherited the expression pattern of the F_1_ progenitor, 22.3% reverted to the F_0_ pattern, 22.3% reverted to the basal expression pattern of non-stressed F_0_ plants. The remaining 18.8% showed new expression patterns. Not unexpectedly, these proportions varied between different genes. Collectively, the results of two studies showed that the altered gene methylation and expression states induced by heavy metal stress are partially heritable between plant generations.

Although the studies described above and many other studies have shown that epigenetic changes play an important role in forming transgenerational stress memory, the mechanisms underlying these epigenetic changes still remain mostly unknown.

## 5. Epigenetic Variability in Nature as an Adaptive Mechanism to Environmental Stress

One of the first studies of epigenetic variability in the natural plant populations used mangrove plants *Laguncularia racemosa* that grow in two neighboring habitats—at the riverside or near a salt marsh [98]. The most visible phenotypic differences between these plants are larger height and tree diameter in those growing at the riverside. Genome-wide analyses of genetic variability by the AFLP (amplified fragment length polymorphism) method, and of epigenetic variability by the MSAP method showed that plants from both habitats are quite similar genetically, but very different epigenetically. In the salt marsh plants, the DNA methylation level is significantly reduced compared with riverside plants.

Invasive populations of the Japanese knotweed (*Fallopia japonica* and two its closely related species *F. sachalinensis* and *F. x bohemica*) occupied a wide range of habitats in Europe and even more different habitats in the USA [99]. Plants from 16 populations that occupied different habitats (marsh, beach, and roadside) showed very low genetic variability. Of ~200 AFLP sites analyzed, only four appeared to be polymorphic, yielding 8 haplotypes that showed differentiation primarily by taxa and, to a lesser degree, by habitats. In contrast, epigenetic variation was much higher. Out of 180 MSAP sites, 19 were polymorphic, creating 129 epigenotypes. Nearly all habitats were epigenetically differentiated. Moreover, this pattern was present not only when the whole collection of Japanese knotweed *sensu lata* was analyzed, but also in the *F. japonica sensu stricto*, the hybrid *F. x bohemica*, or just two the most common hybrid haplotypes E and G. In different habitats, the epigenetic marks differentiated faster than genetic markers. Furthermore, pairwise comparisons showed that epigenetic differentiation occurred independently of genetic differentiation. The question that remained unanswered is whether the high epigenetic variability allowed the invasive plans to succeed in colonizing the new habitats, or different new habitats induced the epigenetic differences.

A study of genetic and epigenetic variation in the offspring of apomictic dandelion (*Taraxacum officinale*) collected in different regions in Europe revealed high levels of local variations and modest levels of heritable variation between different locations [100]. Genetic and epigenetic variations appeared to be significantly correlated, reflecting the dependence of epigenetic variation on the genetic factors. However, a small share of DNA methylation variation appeared to be independent of genetic variation. This epigenetic variation might reflect environment-induced heritable changes in DNA methylation that affect fitness and could, therefore, serve as selectable features in environmental adaptation. Apomictic clones of the same apomictic *T. officinale* lineage collected from different field sites showed heritable differences in flowering time correlated with inheritable differences in DNA methylation patterns [101]. These differences in flowering time were significantly reduced after in vivo DNA demethylation with an inhibitor zebularine. Zebularine did not cause consistent changes in flowering time. Rather, it synchronized the flowering curves between different accessions. Thus, differences in DNA methylation between accessions probably mediate the flowering time divergence. Hence, epigenetic variation may cause heritable phenotypic differences in ecologically relevant traits in natural plant populations. Commonality and heritability of environmentally induced changes in DNA methylation were studied by subjecting multiple accessions of two different apomictic dandelion lineages (*T. alatum* and *T. hemicyclum*) to drought and SA [102]. In both stress treatments, heritable variations of DNA methylation were accumulated across three successive plant generations, indicating a high frequency of spontaneous epimutations. Less evident were the directional effects of environment on DNA methylation. Drought stress induced some accession-specific changes in DNA methylation in the exposed generation but not in the unexposed offspring. By contrast, SA caused increased changes in DNA methylation in the offspring of exposed plants compared with the offspring of non-stressed plants. These changes increased transgenerational epigenetic variation between the individual offspring plants but did not cause predictable epigenetic variants. Therefore, stress-induced heritable changes in DNA methylation are genotype- and context-specific but mostly undirected and not targeted to specific loci.

*Spartina alterniflora* and *Borrichia frutescens* are two plant inhabitants of the Atlantic coastal salt marshes that demonstrate high environment-correlated phenotypic plasticity [103]. These two species represent a contrast in habitat preference. *S. alterniflora* grows in dense monospecific stands in the lower elevations of the marsh subject to daily tidal submergence. It is a native plant species along the entire east coast of the United States and an invasive species worldwide. In the upper, more saline parts of the marsh, the salt-tolerant species *B. frutescens* predominates. Both species inhabit a broad range of environments, with highly variable soil salinities (20 to >100 ppt) and show extreme phenotypic variation in height and the number and size of leaves. The soil is the strongest predictor of these variations accounting for >50% and 20% of the height variations in *S. alterniflora* and *B. frutescens*, respectively. Genetic and epigenetic variations were investigated in different populations and habitats within populations (subpopulations) on a spatial scale that maximized the collection of unique genotypes within the low, intermediate, and high salt habitats [103]. Genome-wide genetic variation in individual plants was studied by an AFLP method, while an MSAP method was used to study the epigenetic variation. A significant genetic variation was found between subpopulations in both species, but it did not correlate with their habitats. High epigenetic variation was found between subpopulations inside each population and between those from different populations. Moreover, a weak but significant effect of habitat type was detected in both species, suggesting consistent epigenetic differentiation to habitat type. However, a significant correlation was also found between genetic and epigenetic variation. In *S. alterniflora*, a weak but significant correlation between epigenetic variation and habitat was found after removing the effect of genetic variation. In contrast, no correlation between genetic variation and habitat was detected when the effect of epigenetic variation was removed. Even stronger epigenetic correlations with habitat were found inside the individual population. In *B. frutescens*, significant correlations between epigenetic variation and habitat were also found, but these correlations were in opposite directions and weak in two populations. Furthermore, weak but significant correlations were detected between genetic variation and habitat in two populations.

In two large dataset of *Arabidopsis thaliana*, Eurasian panel [104] and Swedish panel [105], climate and geographical distance variables explained 7.5% and 1% of the variation in single-nucleotide polymorphisms (SNPs) and 2.5% and 5% of the variation in single methylation site polymorphisms (SMPs), respectfully [106]. The amounts of variation explained by climate independent of geographical distance were two- to threefold less. In the Eurasian panel, 6.5% of the variation in all-cite DMRs (C-DMRs) and 4.7% of the variation in CG-DMRs were explained by climate and geographical distance parameters. In the Swedish panel, climate and space explained 16% of the variation in C-DMRs and 18% of the variation in CG-DMR. In both panels, CHH methylation in transposable elements showed the highest explanatory power, followed by CG-DMRs in gene bodies. These CG-DMRs were enriched for genes involved in responses to abiotic stimuli, reproduction, development, and metabolism. Unlike C-DMRs, CG-DMRs were positively correlated with gene expression when located within their bodies. These data support the previous study that have found a strong association between climatic temperature and TE methylation at CHH sites within and nearby the *CMT2* gene [105]. Similar associations of DNA methylation variation with climate factors in two panels indicate its possible contribution to local climate adaptation.

In a perennial herb *Helleborus foetidus L*., epigenetic diversity was found to be spatially structured, with the epigenetic similarity between plant pairs declining significantly with increasing distance between their habitats [107]. The slopes of similarity–distance regressions for epigenetic markers were considerably steeper than genetic markers, indicating a higher increase in epigenetic variability with increasing geographic distance compared with genetic variability. Only in plant pairs from the same subpopulation, the epigenetic similarity between individual plants was substantially more significant than their genetic similarity, probably due to plants being exposed to the same environmental conditions.

*Lilium bosniacum* is a rare species endemic to the Balkan peninsula. Its typical habitats are rich in nutrients, exposure to sunlight, and located at an altitude from 1200 to 1300 m. However, there is a small population of *L. bosniacum* that grows on serpentine soil, characterized by low amounts of essential nutrients (N, P, K), low Ca/Mg ratio, high concentrations of heavy metals, and water deficit. In a recent paper, genetic, epigenetic, and cytogenetic differences were investigated in three natural populations of *L. bosniacum* from contrasting habitats [108]. The first one (P01) was the unique population growing on serpentine soil, the second (P02)—an alpine population growing under stresses of high altitude, and the third (P03)—a population growing under customary ecological conditions for this species. The harsh conditions of the P01 population include an unusually low altitude, a bare serpentine substrate, the lowest light availability compared with the other two, and harsh continental climate conditions (below-freezing temperature during 3 months, severe and late frosts, and abundant precipitation). The epigenetic differences between populations were much larger than the genetic differences. Genetic distances between populations showed a tendency of separation, but due to high intra-population dispersion and a significant overlap between populations, no strong genetic population structure was observed. In sharp contrast, epigenetic distances showed clear separation of all three populations, suggesting high levels of epigenetic divergence among populations. Indeed, 4.95% of the total genetic variation and 11.72% of the epigenetic variation were attributable to the differences between populations. These findings suggest that habitats strongly affect the epigenetic component of the genome in a convergent manner. Most of the total variation was attributed to the unmethylated state of the epiloci that was associated with unfavorable environmental conditions in the habitats. In total, these data suggest that epigenetic variation might be the first, immediate source of phenotypic variability that plays an important role in responses to stress and provide the necessary time for the natural selection of genetic variants for the long-term stable adaptation to a new environment.

Many tree and shrub species reproduce asexually, resulting in genetically identical offspring plants. Lombardy poplar *Populus nigra* cv. *Italica* Duroi is a cultivated variety of *P. nigra* L. that originated at the beginning of the eighteenth century from one single male mutant tree in central Asia from where it was spread by cuttings through Europe and worldwide [109]. Epigenetic variability seems to be the most plausible mechanism of its adaptation to widely variable climatic conditions without genetic variation. In the 60 vegetative offspring plants (F_1_ generation) of genetically homogenous F_0_ generation Lombardy poplars collected at 37 geographic locations in Europe and Asia, 94 methylation-susceptible epiloci were detected, of which 65 (68%) were polymorphic. In contrast to common genotype, all the 60 epigenotypes were unique. The pairwise differences between individual epigenotypes varied from 2 to 23 epiloci (mean = 17). Significant epigenetic differences were found between countries of F_0_ tree origin, but no correlation between the pairwise epigenetic distances and the geographic distances was found. An unmethylated state was detected in 68 polymorphic methylation-susceptible epiloci. Since all the F_1_ plants were grown under the controlled greenhouse conditions, the observed methylation variation probably was caused by differences in the environment of the donor (F_0_-generation) trees. These data support the view that the environment promotes epigenetic variation that could be inherited between plant generations. The authors could not find direct links between epigenotypes and climate variables. Probably, the number of epiloci in the MSAP analysis used was too small to detect such links robustly.

Scots pine (*Pinus sylvestris* L.) is a long-lived conifer species adapted to a wide range of environmental conditions. Several studies showed low genetic variability between Scots pine populations, but high variation in phenotypic traits related to cold tolerance and photoperiod [110]. The total DNA methylation levels in megagametophyte and embryo showed some variation between populations from northern and southern Finland, but these differences were not statistically significant. Six DNA methyltransferase (*DNMT*) genes belonging to all three families (*MET1*, *CMT*, and *DRM*) were identified and sequenced in Scots pine. Significant variability in *DNMT* gene expression was observed between individual pine trees, but no significant differences were found between the pine populations. In total, the differences observed did not differ from those expected by chance. Of 19 adaptation-related genes studied, significant differences in the expression levels of 11 genes were detected in megagametophytes, and of 8 genes in embryos. In both tissues, gene expression showed a similar correlation with three climate variables: accumulation of growing degree-days, length of the growth season, and average rainfall from May to September. An opposite correlation was observed with the day length during the growth season. The genes whose expression varied significantly between the pine populations showed the strongest correlation with climate variables. Of *DNMT* genes, *MET1-1* and *CMT* showed a strong positive correlation with day length during the growth season and a strong negative correlation with the growth season length and the accumulation of growing degree-days. *MET1-1* showed a negative correlation with average rainfall from May to September. This differential expression of *DNMT* genes may be involved in forming epigenetic memory to allow rapid adaptation to the changing environment, especially in the harsh northern conditions. Altogether, these data suggest that DNA methylation and gene expression variability contributes to Scots pine adaptation to local climate conditions.

The model grass *Brachypodium distachyon* has a small and fully sequenced genome and a wide climatic distribution resulting in phenotypic diversity [111]. Multiple genetically similar accessions from different environments across Turkey were selected and grown alongside with the reference genotype Bd21 in growth chambers under simulated spring or fall conditions of Turkey’s climate. Previous analyses showed that some of these accessions had highly similar genome sequences (by SNP analysis similar to that observed between technical replicates), but were located in different geographical regions. These accessions (BdTRs) were grouped into seven nearly genetically identical “families.” A low-coverage WGBS analysis showed that CG methylation patterns nearly reproduced the known genetic relationships, grouping accessions into previously determined families. The low coverage prevented accurate measurements for non-CG methylation contexts. Differentially methylated regions in the CG context (CG-DMRs) were identified in pairwise comparisons between (i) replicates of the same accession—stochastic differences, (ii) accessions within families—intra-family variability, and (iii) between families—inter-family variability. Epigenetic variability increased with increasing genetic distance—a greater mean number of DMRs, greater magnitude, length, and CG count were identified between families than in intrafamily comparisons. Nevertheless, substantial epigenetic variability was detected within accessions and family groups, which may contribute to heritable phenotypic variation. Statistical modeling showed that most phenotypic variance could be explained by additive genetic effects, except two traits that showed a significant epigenetic component. It was estimated that 10% of the variation in flowering time in spring conditions and 16% of the variation in plant height in fall conditions are due to methylation differences.

## 6. Conclusions

Many studies showed that epigenetic variation could be an important mechanism to adapt to different habitats. Immobile plants succeed in surviving in a changeable and often hostile environment due to high phenotypic plasticity. Plants have evolved sophisticated mechanisms to respond and adapt to various stresses. These mechanisms operate at various time scales, from short-term physiological and metabolic responses to long-term genetic and epigenetic modifications (Figure 1). The short-term mechanisms are essential in the immediate survival of the stressful conditions, while the long-term modifications could be of evolutionary significance in providing a stable molecular basis for phenotypic plasticity to select for a progeny that is more adapted to a permanently changing environment. Many studies showed that such phenotypic plasticity is observed in natural genetically homogenous populations, suggesting its epigenetic nature. There is still no direct evidence of a causal relationship between epigenetic and phenotypic plasticity observed in plants of different habitats or exposed to various stresses.

Nevertheless, the fundamental features of epigenetic signals, such as their important role in the control of gene expression and their stability and heritability in successive plant generations, support the view of epigenetic variations as a unique adaptation mechanism of evolutionary significance [26,112]. Unlike classic mutations, adaptive epigenetic changes can occur at much shorter timescales than those needed to select adaptive mutations. Despite their stability, epigenetic signals are essentially reversible. On the evolutionary time scales, the epigenetically induced adaptive phenotypes would probably be genetically assimilated. However, even considering its short lifetime, epigenetic variability plays an essential role in plant adaptation to a changing environment.

Because of their role in regulating gene expression, epigenetic variations can create the phenotypic differences affecting individual fitness and, therefore, can serve as the material of natural selection. Unlike classic genetic mutations, changes in DNA methylation (epimutations) may occur very rapidly in response to environmental stress and provide potential means to cope with it on a very short time scale. Therefore, DNA methylation could be an evolutionary relevant process even in the absence of transgenerational inheritance. However, concerning long-term adaptation and evolution, the most promising epialleles are those capable of being inherited between generations independent of specific genetic loci. Alternative epialleles established in response to environmental stress could give the plants that possess them an alternative phenotype. The frequency of potentially favorable epialleles will be higher in subsequent generations. Even this increased phenotypic plasticity per se could be advantageous in rapidly changing environments, since it may help adapt to new environments and provide a more expansive habitat. Based on their dependence on genetic factors, epigenetic variants (epialleles) are classified into three types: (i) “obligate” that display a complete dependency on a genetic feature, (ii) “facilitated” that arise as a result of a genetic feature but could be maintained further in its absence, and (iii) “pure” that do not depend on any genetic features [113]. Most of the epialleles known today are obligate or facilitated, but a few pure epialleles were also found. Thus, significant phenotypic variation can arise due to changes in DNA methylation that can potentially be inherited by subsequent generations. However, the epigenetic variation depends, to a very significant extent, on the underlying genetic variation, and these two types of variation should be analyzed together.

## Figures and Tables

**Figure 1 ijms-21-07457-f001:**
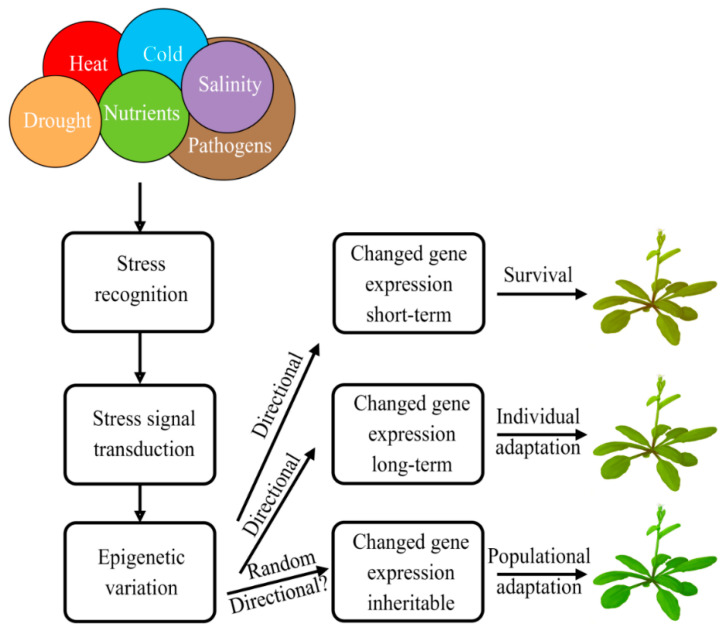
Epigenetic mechanisms of the short-term and long-term plant adaptation to environmental stresses.

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
