# Peer review of "Epigenetic Mechanisms of Plant Adaptation to Biotic and Abiotic Stresses"

_ijms, 2020, doi:10.3390/ijms21207457_

Round 1

Reviewer 1 Report

This review paper focuses on the epigenetic regulation of plant response to biotic and abiotic stresses. The topic has been pretty much covered, but frankly, this manuscript is hard to read as it is too long and includes a lot of detailed summary of literatures that seems unnecessary. Also, there are some relevant studies, which might be of interest to the readers, are lacking, for example, the epigenetic regulation during fruit development and storage, on which a mature model has been proposed in tomato. Another one is RNA modification, which is an emerging area adding to our knowledge of epigenetics. I understand that RNA epigenetics is still in the early stage and does not have broad impact at this time, but I encourage the authors at least to mention this in the manuscript. To conclude, while I find this review paper interesting to me, I strongly feel that the length needs to be shortened, and a concise version of this manuscript should be seriously considered to be published in IJMS.

Author Response

Dear Reviewer,

Thank you very much for your suggestions. We have deleted multiple fragments of the text containing the details that seemed unnecessary. These deletions are too numerous to make a list of them but they are visible in the revised version of the manuscript due to our use of the "Track Changes" function of the Microsoft Word. As concerning relevant studies that are lacking, we did not intend this paper as a comprehensive review of the whole field of ecological epigenetics. There are quite a number of published reviews on this topic (some of them are referenced in the last paragraph of the Introduction). That is why we have chosen to review in more detail selected papers that were mostly published recently and contain robust data essentially contributing to the knowledge of the epigenetic mechanisms of adaptation. Of course, we do understand that this selection is somewhat subjective and reflects our personal view of the problem. We are very much interested in some new aspects of plant and animal epigenetics, such as adenine methylation of DNA and epitranscriptomes. However, these aspects are still in their infancy. As a matter of fact, we are not even sure that these types of DNA and RNA modifications could be regarded as epigenetic sensu stricto. That is why we have chosen not to discuss them in this paper.    

Reviewer 2 Report

This review presents epigenetic regulations allowing plants to adapt to environmental
stresses. Although the topic is interesting, I think that modifications could improve this
review. Please find my comments below.
- The abstract should clearly answer these questions:
o What is the goal of this review?
o What is the novel aspect presented in this review that was not already
described in previous reviews?
- I think that more than one figure should be included in the review. Additional
figures and tables would greatly enhance the quality of the review and would help
the reader.
- The introduction should also introduce the concepts of stress perception and
signaling.
- Subheadings should be added within each section in order to guide the reader.
- This review seems like a patchwork made of different detailed summaries coming
from selected publications.
o Lines 223-250: only one reference was cited in this whole paragraph. This
seems a complete summary of one article.
o Lines 258-280: only one reference was cited in this whole paragraph. This
seems a complete summary of one article.
o Lines 283-335: only one reference was cited in this whole paragraph. This
seems a complete summary of one article.
o Lines 419-457: only one reference was cited in this whole paragraph. This
seems a complete summary of one article.

Connections and transitions should be added between the different paragraphs
and sections to guide the reader.
- The detail level should be adapted for the review style I think, with a focus on
results that support the goal of this particular review.
- Lines 184-186: The grey color should be removed.
- Line 80: “other epigenetic pathways”, please precise other than which pathway.
In summary, I think that extensive changes are required and I do not recommend the
publication of this review in the current form.

Author Response

Dear Reviewer,

Thank you very much for your suggestions. We have revised the manuscript globally. Since the changes were numerous we could not make a list of them but these changes are readily visible thanks to the “Track changes” function of Microsoft Word that we have used. The volume of the revised version is substantially less than the original manuscript but still, it is too long to add new figures and/or tables. We have chosen to explain our intentions concerning the goal of the review and its new aspects in the last paragraph of the Introduction instead of including them in the Abstract that is strictly limited in volume by the journal instructions. Relevant descriptions of the stress perception and signaling are present in respective parts of the main text. We have not included this information into the Introduction because it is rather different for various kinds of stress. We have added subheadings within sections. We have deleted multiple parts of the original text that contained excessive summaries of individual studies.

Round 2

Reviewer 1 Report

My concerns have been addressed and now I support the publication of this manuscript at IJMS.

Reviewer 2 Report

The manuscript has been greatly improved and I do not have further suggestions.